# Synthesize Privacy-Preserving High-Resolution Images via Private Textual Intermediaries

**Haoxiang Wang**
Peking University
wanghaoxiang@stu.pku.edu.cn

**Zinan Lin**
Microsoft Research
zinanlin@microsoft.com

**Da Yu**
Google Research
dayuwork@google.com

**Huishuai Zhang**$^{†}$
Wangxuan Institute of Computer Technology, Peking University
State Key Laboratory of General Artificial Intelligence
zhanghuishuai@pku.edu.cn

## Abstract

Generating high-fidelity, differentially private (DP) synthetic images offers a promising route to share and analyze sensitive visual data without compromising individual privacy. However, existing DP image synthesis methods struggle to produce high-resolution outputs that faithfully capture the structure of the original data. In this paper, we introduce a novel method, referred to as *Synthesis via Private Textual Intermediaries* (*SPTI*), that can generate high-resolution DP images with easy adoptions. The key idea is to shift the challenge of DP image synthesis from the image domain to the text domain by leveraging state-of-the-art DP text generation methods. *SPTI* first summarizes each private image into a concise textual description using image-to-text models, then applies a modified Private Evolution algorithm to generate DP text, and finally reconstructs images using text-to-image models. Notably, *SPTI* requires no model training, only inferences with off-the-shelf models. Given a private dataset, *SPTI* produces synthetic images of substantially higher quality than prior DP approaches. On the LSUN Bedroom dataset, *SPTI* attains an FID = 26.71 under $\epsilon = 1.0$, improving over Private Evolution's FID of 40.36. Similarly, on MM-CelebA-HQ, *SPTI* achieves an FID = 33.27 at $\epsilon = 1.0$, compared to 57.01 from DP fine-tuning baselines. Overall, our results demonstrate that *Synthesis via Private Textual Intermediaries* provides a resource-efficient and proprietary-model-compatible framework for generating high-resolution DP synthetic images, greatly expanding access to private visual datasets. Our code release: https://github.com/MarkGodrick/SPTI

## 1 Introduction

The past few years have seen an explosion in the capabilities of state-of-the-art generative models. Large diffusion and autoregressive models can produce photorealistic images, coherent long-form text, and convincing multi-modal outputs with little more than a prompt [36, 17, 30, 43, 29, 56, 48, 47]. However, when these powerful public models are applied to domain- or user-specific tasks, some form of adaptation is often required, whether it be expensive fine-tuning on private data or in-context learning with user examples. At the same time, the use of private or sensitive data raises serious privacy concerns: direct fine-tuning of a model on proprietary images or user uploads risks memorization and potential leakage of personal information [7, 34, 49, 55].

---

$^{†}$Corresponding author.

39th Conference on Neural Information Processing Systems (NeurIPS 2025).

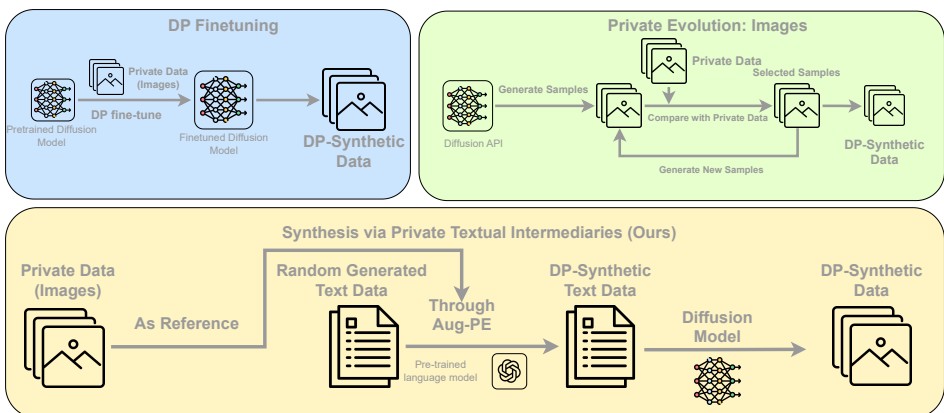

Figure 1: Overview of the *Synthesis via Private Textual Intermediaries* (*SPTI*) framework for differentially private (DP) synthetic data generation. **Top left**: *DP fine-tuning framework.* A pretrained model is fine-tuned on private data under DP constraints, and the resulting model is used to generate DP synthetic data. **Top right**: *Private Evolution (PE) on images.* This method begins by randomly generating candidate samples, which are then compared to private data. Samples are selected based on a voting mechanism and further perturbed to produce a new generation of samples. **Bottom**: Synthesis via Private Textual Intermediaries *(*SPTI*) framework.* Private image data is served as reference. A modified Augmented Private Evolution (Aug-PE) method [53] is then applied to generate DP synthetic text data, which is subsequently transformed into DP synthetic images using a diffusion model API.

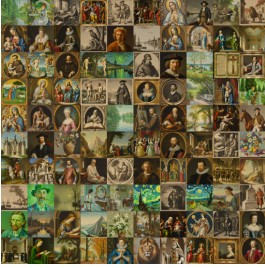

Figure 2: DP-synthetic images from dataset **European Art** ($\epsilon = 1.0$).

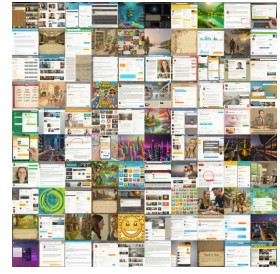

Figure 3: DP-synthetic images from dataset **Wave-ui-25k** ($\epsilon = 1.0$).

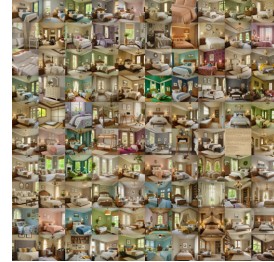

Figure 4: DP-synthetic images from dataset **LSUN Bedroom** ($\epsilon = 1.0$).

Broadly speaking, there are two strategies to harness private data under formal differential privacy guarantees [10]. The more direct approach is *differentially private (DP) fine-tuning* [57, 25, 6, 23, 35, 31, 8], in which gradient updates are clipped and noised (e.g., via DP-SGD [1]). While conceptually straightforward, this method demands substantial expertise and computational resources. For instance, fine-tuning a state-of-the-art model such as DeepSeek V3 [29] requires hundreds of GPUs, not to mention the engineering complexity of per-sample gradient clipping and noise calibration of DP-SGD [15]. Moreover, many proprietary foundation models do not even permit user fine-tuning, effectively precluding this route.

An alternative is to generate a *DP synthetic dataset* that mimics the distribution of the private data [27, 53, 28, 12] and then use that synthetic data for downstream adaptation—either by fine-tuning or by providing examples in-context. This "generate-then-adapt" paradigm sidesteps the need to directly privatize a large model but poses its own challenges: how can one synthesize high-fidelity images (or other modalities) that both faithfully capture the structure of the original data and satisfy strong DP guarantees, all without incurring substantial compute overhead?

In this work, we introduce a *Synthesis via Private Textual Intermediaries* framework that leverages the remarkable text understanding of modern models as a bridge between private data and high-quality generation. Our key insight is twofold:

1. **Text generation is one of the most successful modalities for AI, so is the private text generation.** Private text generation can achieve similar performance to directly differentially private fine-tuning LLMs [53, 59], and text models can be privately adapted far more easily than their image counterparts.

2. **Text unifies modalities.** In modern models, natural language often serves as a universal interface, with a wide range of off-the-shelf models available for both generating images, video, audio, and other modalities from text [43, 40, 30], and for generating text from these modalities [47, 18].

Building on these observations, we propose *Synthesis via Private Textual Intermediaries* (*SPTI*) (Figure 1). Given a private image collection, *SPTI* first converts the images into concise textual summaries using a standard captioning model. It then adapts *Private Evolution* [27, 53, 28, 50], a DP synthetic data algorithm that only requires model inference, to generate DP-sanitized captions that reflect the private caption distribution. Finally, it uses an off-the-shelf text-to-image models to sample candidate images conditioned on these DP captions. Notably, all steps in *SPTI* rely solely on the inference APIs of existing models without any additional training, allowing *SPTI* to leverage state-of-the-art models behind proprietary APIs—something that DP fine-tuning approaches cannot achieve.

We evaluate *SPTI* on standard benchmarks. On LSUN Bedroom, *SPTI* achieves FID = 26.71, markedly improving over Private Evolution's prior best of 40.36 for $\epsilon = 1$. On MM-CelebA-HQ, it attains FID = 33.27 versus 57.01 for DP fine-tuning baselines for $\epsilon = 1$. In both cases, *SPTI* generates high-resolution synthetic images with significantly better quality than either cost-intensive DP-SGD approaches [43] or previous API-based approaches [27]. Figures 2, 3, 4 show generated images from *SPTI*.

Our contributions are as follows:

- Conceptually, we introduce *Synthesis via Private Textual Intermediaries*, a novel framework that uses text as an intermediate representation to bridge off-the-shelf multimodal LLMs and achieve privacy-preserving, high-resolution image generation by leveraging state-of-the-art DP text generation methods.

- Technically, we develop a novel Private Evolution voting mechanism that privately select text descriptions by voting in image representation space. It achieves strong privacy while ensuring strong resemblance to the private image data.

- Empirically, we demonstrate that *SPTI* generates high-quality, high-resolution images across diverse datasets, and can readily incorporate future advances in multimodal generation. Extensive ablation studies validate the effect of each components of the algorithm in achieving state-of-the-art performance.

The rest of the paper is organized as follows. In Section 2 we review background and preliminaries; Section 3 details the *SPTI* method; Section 4 presents experimental results and analyses; Section 5 discusses related work; Section 6 concludes with a discussion of future directions; and Section 7 faithfully describes the potential limitations of the proposed method.

## 2 Preliminaries: Differential Privacy and DP Synthetic Data via APIs

**Differential Privacy.** One mechanism $M$ is said to be $(\epsilon, \delta)$-DP if for any two neighboring datasets $D$ and $D'$ which differ in a single entry and for any observation set $S$ of outputs of $M$, one has

$$\Pr\big(M(D) \in S\big) \leq e^\epsilon \Pr\big(M(D') \in S\big) + \delta.$$

The intuition of $(\epsilon, \delta)$-DP requires that any single sample cannot influence the mechanism's output too much.

**DP Synthetic Data via APIs.** The DP synthetic data problem is formulated as DP Wasserstein Approximation in Lin et al. [27]. Given a private dataset $\mathcal{D}_{priv} = \{x_i : i \in [N_{priv}]\}$ with $N_{priv}$ samples (e.g., images), the goal is to design an $(\epsilon, \delta)$-DP algorithm $M$ that outputs a synthetic dataset

$\mathcal{D}_{syn} = \{x'_i : i \in [N_{syn}]\}$ with $N_{syn}$ samples, where the Wasserstein p-distance $W_p(\mathcal{D}_{priv}, \mathcal{D}_{syn})$ is minimized, w.r.t. the distance function $d(\cdot, \cdot)$ and some $p \geq 1$.

**Private Evolution (PE).** [27] is a recently proposed differentially private synthetic data generation that relies solely on API access to off-the-shelf models. The PE algorithm consists of following steps:

- **Random Initialization.** Call RANDOM_API to draw an initial set of synthetic candidates from the foundation model.

- **Repeat the following routine several times**

  1. **Private Voting.** Using the private dataset, each real sample "votes" for its nearest synthetic candidate under the embedding network, producing a (noisy) histogram via the DP voting function.

  2. **Resampling.** Draw a new batch of synthetic points by sampling from this private histogram.

  3. **Variation.** For each drawn point, invoke VARIATION_API to produce novel samples that are semantically similar to the selected candidate (e.g., variations of an object in an image).

This procedure can generate synthetic data whose distribution closely matches that of the private dataset while controlling privacy leakage using a private histogram algorithm.

# 3 Method: High Quality Image Synthesis via Private Textual Intermediaries

Before describing our method, we first outline the high-level motivation. The goal of differentially private (DP) synthetic data generation is to mimic the distribution of private data, no matter how rare or complex the private data are, e.g., high-resolution or UI-like images, while preserving privacy. However, directly synthesizing such images by using Private Evolution [27] poses two key challenges. First, rare cases are difficult to generate if they are far from typical data distribution that the model has learned on. Second, high-resolution images require the model to correctly render all fine-grained details across a large output space, which is particularly challenging for generation models.

To address these issues, we shift the problem from the image domain to the text domain by introducing textual intermediaries. Text representations are semantically rich yet lower-dimensional, making them significantly more amenable to efficient DP synthesis. Recent advances have shown that privately trained language models [59] or DP synthetic text [53] can achieve strong utility while satisfying DP. Moreover, text serves as a universal interface for multimodal generation: a single caption can condition powerful off-the-shelf models to generate images, videos, audio, and more. This makes our approach have the potential to be broadly applicable across modalities, though we focus on image synthesis in this work.

We propose *Synthesis via Private Textual Intermediaries* (*SPTI*), a novel framework that leverages this insight. *SPTI* first converts private images into captions, applies our newly-designed Private Evolution (PE) algorithm to these descriptions under a DP guarantee, and then synthesizes images using a text-to-image generator. This design ensures strong privacy while maintaining high utility. An overview is shown in Figure 1.

## 3.1 The Design of *Synthesis via Private Textual Intermediaries*

Our *SPTI* consists of three stages: (1) using image-to-text model to caption private images to obtain text descriptions, (2) applying Private Evolution to the text data under DP constraints, and (3) using text-to-image model to generate synthetic images using the evolved text. The process is outlined in Algorithm 1.

**Images to Texts.**    Given a private image dataset $\mathcal{D}$, we begin by captioning each image into a textual description using an off-the-shelf image captioning model. This results in a corpus of private text data $\mathcal{T}$, which serves as a proxy for the original image data during subsequent privacy-preserving operations.

**Private Evolution in Text Space.**    Augmented Private Evolution (Aug-PE) [53] is an iterative, population-based method for generating synthetic text data under DP constraints. We modify it by replacing its voting mechanism with a new voting strategy. At each iteration, a pool of candidate texts is generated, evaluated, and selected. For selection, we assign each candidate a probability using Image Voting process, which is described with detail in Section 3.2. We then select the candidates by this probability to produce the next generation. This process preserves privacy while exploring the text space efficiently. We provide a detailed version of this modified algorithm 2 in Appendix A.

**Evolved Texts to Images.**    Once we obtain the evolved synthetic text set $\mathcal{T}'$, we use a high-quality text-to-image diffusion model (e.g., state-of-the-art open-source models [44] or commercial APIs) to convert each text back into an image. Since the evolution occurred entirely in the text domain, the resulting images are inherently DP-compliant due to post processing properties of DP.

**RANDOM_API and VARIATION_API.**    Our method follows **Aug-PE** [53] method, also applying `RANDOM_API` and `VARIATION_API` in our algorithm `Aug_PE_Image_Voting`. For `RANDOM_API`, we prompt the LLMs to generate random image captions from scratch, denoted as `RANDOM_API`$(N)$. This means `RANDOM_API` will generate $N$ samples. For `VARIATION_API`, we also prompt the LLMs to generate variations of the given image captions, denoted as `VARIATION_API`$(\mathcal{C})$. This means `VARIATION_API` will generate variants of equal quantity from text data $\mathcal{C}$.

---

**Algorithm 1** *SPTI*: Privately Synthesize High-Resolution Images via *Synthesis via Private Textual Intermediaries*

---

**Require:** Private image dataset $\mathcal{D}$, `Aug_PE_Image_Voting`
**Ensure:** Synthetic images $\mathcal{D}'$
 1: Convert $\mathcal{D}$ to text descriptions $\mathcal{T}$ using a captioning model
 2: Apply Private Evolution: $\mathcal{T}' = $ `Aug_PE_Image_Voting`$(\mathcal{T})$
 3: Generate images $\mathcal{D}'$ from $\mathcal{T}'$ using a text-to-image diffusion model
 4: **return** $\mathcal{D}'$

---

### 3.2    Private Evolution in Textual Space with Voting by Image Representations

Running Aug-PE on image captions can generate synthetic captions that are similar to those of the original private images in the text domain. However, the final text-to-image models work in a random and complex way. As a result, even accurate captions in the text domain may not necessarily produce images that closely match the original private ones in the image domain.

Based on this insight, we propose **Image Voting** (Algorithm 2), a cross-modal selection mechanism integrated into the PE process. Rather than selecting candidate texts solely based on textual similarity, we first generate images from the candidate texts using the text-to-image model. These synthesized images are then embedded using an image encoder (e.g., Inception-v3 [46, 39]), along with the private images.

For each private image embedding, we identify its nearest neighbor among the generated image embeddings (e.g., via Euclidean distance). The corresponding synthetic text associated with the nearest image is assigned one vote. After accumulating votes across all private images and adding noise, we normalize the vote counts into a probability distribution over candidate texts, which is then used to sample the next generation. This reverse-nearest-neighbor approach prioritizes synthetic texts that produce images broadly aligned with the private dataset. Additionally, the **Image Voting** process notably **requires no textual reference**, as it directly leverages the private images themselves as guidance during generation.

### 3.3    Privacy Analysis

We analyze the privacy guarantees of our proposed *SPTI* framework, which synthesizes image data under differential privacy (DP) by operating in the text space. Our goal is to ensure that the final synthetic images $\mathcal{D}'$ are generated through a process that satisfies $(\varepsilon, \delta)$-DP with respect to the private dataset $\mathcal{D}$.

**Privacy-Critical Step.** The only step in the *SPTI* pipeline that accesses the private data is the Private Evolution module, specifically through the similarity-based voting mechanism in algorithm `Aug_PE_Image_Voting` (Algorithm 2). This module operates on embeddings of the private image dataset $\mathcal{D}$ and is used to guide the sampling of synthetic text candidates. Thus, our privacy analysis focuses on this step.

**Voting Mechanism.** For each private image embedding, we identify its nearest neighbor among the generated image embeddings , and increment a vote count histogram $H$ indexed by the candidate text that produced the nearest image. This step corresponds to a histogram query over the private data.

To ensure differential privacy, we add Gaussian noise $\mathcal{N}(0, \sigma^2 \mathbf{I})$ to the vote histogram $H$ , which is equivalent to applying the *Gaussian mechanism* to a function with bounded sensitivity.

**Sensitivity Analysis.** The histogram query has $L_2$ sensitivity at most 1, since each private image contributes a vote to only one candidate (i.e., changing a single image can affect the histogram by at most 1 in one coordinate). This satisfies the precondition for the Gaussian mechanism.

According to the Gaussian mechanism [4], adding noise of variance $\sigma^2$ per coordinate to a function with $L_2$ sensitivity 1 ensures $(\varepsilon, \delta)$-DP, provided:

$$\Phi\left(\frac{1}{2\sigma} - \varepsilon\sigma\right) - e^\varepsilon \Phi\left(-\frac{1}{2\sigma} - \varepsilon\sigma\right) \leq \delta.$$

**Privacy Composition.** According to the adaptive composition theorem of Gaussian mechanisms [9], applying the above Gaussian mechanism across $G$ iterations satisfy $(\varepsilon, \delta)$-DP, provided:

$$\Phi\left(\frac{\sqrt{G}}{2\sigma} - \frac{\varepsilon\sigma}{\sqrt{G}}\right) - e^\varepsilon \Phi\left(-\frac{\sqrt{G}}{2\sigma} - \frac{\varepsilon\sigma}{\sqrt{G}}\right) \leq \delta. \tag{1}$$

**Post-processing Immunity.** All downstream steps in *SPTI*—including text mutation and image generation via a fixed diffusion model—depend solely on the privatized output of the voting mechanism. By the *post-processing property* of DP, these steps incur no additional privacy cost.

**Overall Guarantee.** Hence, *SPTI* satisfies $(\varepsilon, \delta)$-differential privacy with respect to the private image dataset $\mathcal{D}$, provided that the noise scale $\sigma$ is properly chosen to satisfy Eq. 1.

## 4 Experiments

In this section, we evaluate the effectiveness of our proposed method through comprehensive experiments. Section 4.1 describes our experimental setup. Section 4.2 and 4.3 present the main results, where we compare our approach against state-of-the-art methods across multiple benchmarks to demonstrate its superior performance. Section 4.4 includes ablation studies analyzing the impact of the image voting mechanism and different model APIs. Our findings show that the image voting technique consistently improves performance, while the quality of the synthetic data remains robust across different language and diffusion model APIs. We also represent extended ablation study in Appendix C.

### 4.1 Experiment Setup

**Datasets.** To mitigate concerns that existing benchmarks may have been included in the pretraining data of diffusion models, we carefully select only essential benchmarks with known publication dates, prioritizing those that were introduced recently. Additionally, we create a new dataset by extracting data from the open-source Blender movie **Sprite Fright**. We evaluate our *SPTI* method on six datasets: **LSUN Bedroom** [58], **Cat** [27], **European Art** [41], **Wave-ui-25k** [61], **MM-CelebA-HQ** [51, 52, 33, 21, 24], and our newly constructed **Sprite Fright**. These datasets are selected because each represents a realistic, privacy-sensitive scenario. For instance, **LSUN Bedroom** captures aspects of an individual's private living space, while **Wave-ui-25k** consists of screenshots that may reveal personal information from electronic devices.

**Baselines.** We compare our method with two state-of-the-art DP baselines:

- **Private Evolution** [26]: A recently proposed DP image synthesis method that operates in the image embedding space via private evolutionary strategies.
- **DP Fine-tuning** [32]: This method fine-tunes a diffusion model on private data under DP constraints using DP-SGD, allowing direct sample generation while preserving privacy.

**Models.** Our pipeline requires a captioning model, a large language model (LLM), and a diffusion model. For the caption model, we use **GPT-4o-mini** [38] and **Qwen-VL-Max** [3]. Our experiments show these two models produce similar performances, and detailed results are presented in Section 4.4. For the LLM and diffusion model, we use **Meta-Llama-3-8B-Instruct** [2] for text generation and **SDXL-Turbo** [44] for image synthesis. Additional comparisons with newer and more advanced models are discussed in our ablation study in Section 4.4.

**Evaluation Metrics.** To evaluate the quality of our differentially private synthetic samples, we use the **Fréchet Inception Distance** [16] (FID) between the original private dataset and the synthetic one. Lower FID scores indicate better alignment between the two distributions in terms of visual features.

## 4.2 Evaluation

**FID score**. We compare *SPTI* with two state-of-the-art approaches: **Private Evolution** and **DP fine-tuning**, across multiple datasets. The results are presented in Table 1 and Table 2. For a fair comparison, we use a Latent Diffusion Model (LDM) [42, 5] pretrained on LAION-400M [45] as the backbone for all methods. The experimental results demonstrate the superior performance of *SPTI*. For more details on the experimental settings, please refer to Appendix E.

**Voted samples and variants.** In each iteration, our results generate two categories of samples. One can be categorized as **voted** samples selected from previous generation, while the others can be viewed as variants generated from these samples, which we call **variants**. During experiment evaluation, we often find voted samples perform better on **FID** evaluation, which can be seen in Figures 5, 6. For consistency, we present all our experiment results using FID calculated from voted samples. For more figures on experiment results, please refer to Appendix B.1.

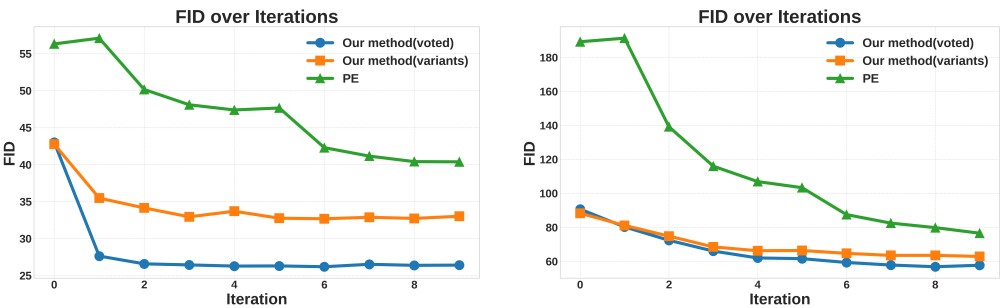

Figure 5: Experiment results on **LSUN Bedroom** dataset

Figure 6: Experiment results on **European Art** dataset

## 4.3 Downstream Task

To comprehensively evaluate the generalization and transferability of the representations learned by our proposed *SPTI*, we conduct a binary classification task on the CelebA dataset. We select the attribute `Wearing_Lipstick` as the target label, and approximately $53\%$ of the samples are without lipstick and $47\%$ are with lipstick. We generate 2,000, 4,000, 6,000, 8,000, and 10,000 samples using *SPTI*, and PE, respectively, and compare their accuracy to the model trained by the corresponding number of ground-truth training samples. A **WRN-40-4** model is then trained on each generated data samples under identical training settings, and the test accuracy is evaluated using the ground-truth labels.

The test accuracy in Figure 7 shows that *SPTI* is more effective than PE method. This could be due to the higher diversity of data generated by *SPTI* compared to that of PE (see Figures 36, 37). We also

|  |  | $\varepsilon = 10$ | $\varepsilon = 5$ | $\varepsilon = 1$ |
|---|---|---|---|---|
| LSUN Bedroom | *SPTI* (ours) | **25.88** | **25.87** | **26.39** |
|  | PE image | 41.72 | 41.08 | 40.36 |
|  | DP-finetune | 31.28 | 31.34 | 31.76 |
| Cat | *SPTI* (ours) | 101.57 | 102.31 | 106.15 |
|  | PE image | **51.00** | **51.29** | **64.62** |
|  | DP-finetune | 148.05 | 148.46 | 148.75 |
| European Art | *SPTI* (ours) | **41.42** | **42.71** | **57.64** |
|  | PE image | 76.25 | 74.41 | 76.50 |
|  | DP-finetune | 61.10 | 61.82 | 63.97 |
| Wave-ui-25k | *SPTI* (ours) | **20.16** | **22.53** | **35.18** |
|  | PE image | 39.28 | 48.95 | 50.45 |
|  | DP-finetune | 49.84 | 52.09 | 62.08 |
| Sprite Fright | *SPTI* (ours) | **142.48** | **141.49** | **157.31** |
|  | PE image | 195.58 | 181.77 | 197.13 |
|  | DP-finetune | 148.19 | 151.94 | 161.25 |

Table 1: **FID** values (lower is better) across multiple datasets to compare three different DP methods: *SPTI*, PE Image, and DP-finetune.

| Method | model | $\varepsilon = 10$ | $\varepsilon = 1$ |
|---|---|---|---|
| *SPTI* (our method) | sdxl-turbo | 26.02 | 34.17 |
|  | LDM pretrained model | **16.59** | **33.41** |
| PE | sdxl-turbo | 53.32 | 50.45 |
| DP-finetune | LDM pretrained model | 50.13 | 57.01 |

Table 2: **FID** values (lower is better) on the MM-Celeba-HQ dataset to compare different DP generation methods: *SPTI*, PE Image, and DP-finetune.

give results from models trained with ground-truth training samples to show that our selected number of samples is sufficient to train a good binary classifier. Additionally, we provide model performance during training and examples of generated samples in Appendix B.3.

## 4.4 Ablation Study

We conduct a series of ablation studies to investigate the contribution of each component of our proposed method. Specifically, we analyze the impact of **image voting** in the main text, while additional studies on hyperparameters for the LLM and diffusion APIs are provided in appendix due to space constraints. All experiments are conducted under the same conditions described in Section 4.1.

### 4.4.1 SPTI Without Image Voting

To evaluate the effect of **image voting**, we substitute it with a baseline approach and present the comparative results in Table 3. Specifically, we compare our image voting strategy with the original Aug-PE method, where synthetic text is voted using generated private text data rather than the original private image data. We observe a significant improvement in FID, suggesting that image-based voting

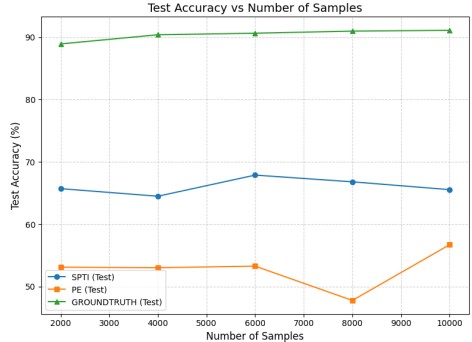

Figure 7: Test Accuracy on **CelebA** dataset with different training samples. *SPTI* achieves better accuracy under the same configuration because SPTI generates samples with more diversity than PE method. The experiment results demonstrate that our method produces data with sufficient fidelity, enabling the model to effectively learn from the generated samples.

plays an important role in enhancing the quality of the generated data. For more figures on experiment results, please refer to Appendix B.2.

| | | $\varepsilon = 10$ | $\varepsilon = 5$ | $\varepsilon = 1$ |
|---|---|---|---|---|
| LSUN bedroom | **Image Voting** | **27.92** | **29.50** | **38.00** |
| | Text Voting | 39.82 | 37.83 | 41.17 |
| Cat | **Image Voting** | **101.57** | **102.31** | **106.15** |
| | Text Voting | 110.51 | 106.07 | 106.02 |
| Europeart | **Image Voting** | **46.69** | **48.45** | **64.04** |
| | Text Voting | 74.37 | 74.04 | 74.91 |
| Wave-ui-25k | **Image Voting** | **34.39** | **42.02** | **70.87** |
| | Text Voting | 92.53 | 92.11 | 95.77 |

Table 3: **FID** values (lower is better) across multiple datasets to compare the *SPTI* method with Image Voting and that with Text Voting.

## 5   Related Work

Differentially private (DP) synthetic data generation has been studied extensively. Early work focused on query-based and statistical approaches, such as the Multiplicative Weights Exponential Mechanism (MWEM) algorithm, which iteratively measures selected queries under DP and synthesizes data to match those noisy answers [14]. PrivBayes learns a Bayesian network with DP noise added to its parameters, then samples synthetic records from the privatized network [60]. While these methods provide provable guarantees, they often struggle with high-dimensional data due to the curse of dimensionality.

The advent of deep generative models enabled DP-GANs and VAEs. Xie et al. [54] apply DP-SGD to GAN training to bound privacy leakage, and PATE-GAN uses the Private Aggregation of Teacher Ensembles framework to train a generator with DP guarantees [20]. Although these models can capture complex distributions, they incur utility loss from noise and training instability.

Recently, diffusion (score-based) models have shown promise under DP. Ghalebikesabi et al. [11] privately fine-tune a pre-trained diffusion model on sensitive images, achieving state-of-the-art FID scores on CIFAR-10 and medical imaging benchmarks. This demonstrates that modern diffusion architectures can yield high-quality private synthetic data. Liu et al. [31] proposed DP-LDM bench-

mark to train latent diffusion models for DP image generation. This provided an outstanding baseline to compare with our method.

A parallel line of work investigates inference-only, API-based DP synthesis. Lin et al. [27] introduce Private Evolution, which injects noise during generation from a frozen model, matching or surpassing retrained approaches on image benchmarks. Xie et al. [53] extend this idea to text, querying a large language model's API with augmented DP sampling to produce synthetic text with strong utility. Lin et al. [28] further extend Private Evolution by incorporating non-neural-network data synthesizers, such as computer graphics tools. This approach delivers better results in domains where suitable pre-trained models are unavailable and unlocks the potential of powerful non-neural-network data synthesizers for DP data synthesis. Google has also demonstrated an inference-only LLM mechanism for DP data generation [13].

A concurrent study also explores generating DP synthetic images by first producing text captions of private images [22]. Their hierarchical DP fine-tuning approach trains a model to generate privatized album descriptions and then photo descriptions conditioned on those. In contrast, we adopt the PE method [53], which requires only API access to foundation models, and further specialize it by using embeddings of the generated images, rather than text captions, during the voting process.

## 6    Conclusion

We have presented *Synthesis via Private Textual Intermediaries* (*SPTI*), a novel framework for differentially private (DP) image synthesis by leveraging existing powerful image-to-text and text-to-image generation models. By first converting each private image into a concise textual summary via off-the-shelf image-to-text models, then applying a modified Private Evolution algorithm to guarantee $(\epsilon, \delta)$-DP on the text, and finally reconstructing high-resolution images with state-of-the-art text-to-image systems, *SPTI* sidesteps the challenges of training DP image generators. Importantly, our approach requires no additional model training, making it both resource-efficient and compatible with proprietary, API-access-only models. Empirical results demonstrate significant improvement over existing methods. These findings validate that shifting the DP burden to the text domain enables the generation of high-fidelity high-resolution synthetic images under strict privacy guarantees, offering a practical path toward sharing and analyzing sensitive visual datasets.

## 7    Limitations

**Domain Generalization**    *Synthesis via Private Textual Intermediaries*' performance is intrinsically tied to the coverage and robustness of the underlying image-to-text and text-to-image models. Although multimodal large language models are continually improving, they still underrepresent certain data domains (e.g., specialized medical imagery, remote-sensing data, niche artistic styles). In such underrepresented or out-of-distribution settings, *SPTI* may fail to capture critical visual features or produce coherent synthesis. Moreover, biases and domain shifts in the pre-trained models can further degrade image fidelity and diversity.

**Computational Overhead**    Our full data generation pipeline involves two major components: an 8B-parameter large language model (LLM) and a 3.5B-parameter diffusion model. Under our current implementation, a complete data generation run takes approximately 19 hours on **a single NVIDIA A800**, or 15.5 hours on **2×NVIDIA A100 GPUs**, with a peak memory requirement of about 70 GB.

In comparison, the baseline PE method [27]uses only a single 3.5B model, requires no LLM, and completes in about 7 hours on **a single NVIDIA A100 GPU**, with a peak memory requirement of about 35 GB. While our method has higher compute demands, it introduces a semantically enriched generation process through language-guided synthesis, which we show leads to superior data diversity and downstream performance.

We believe the additional cost is justified by the significant gains in quality and generalization, and we note that our pipeline can be modularly optimized (e.g., via model distillation or prompt caching) in future work to reduce the runtime.

As for API usage, caption process will take 250M tokens for 10,000 images. During data generation, LLM will process 7.3M tokens during each iteration, and diffusion model will process 5M tokens in each iteration. A total of 10 iterations will count for 123M tokens.

## Acknowledgement

The authors would like to thank the anonymous reviewers for their valuable feedback and suggestions. The work of Haoxiang Wang and Huishuai Zhang was supported in part by National Natural Science Foundation of China (Grant No. 62576015).

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

# A  Details on our `Aug_PE_Image_Voting` algorithm

We present the pseudo-code for the `Aug_PE_Image_Voting` algorithm. While it may appear that the generated private text $\mathcal{T}$ is unused in Algorithm 2, this is by design. Image voting serves solely as a quality enhancement technique within our pipeline. In contrast, the original text voting mechanism explicitly requires the generated private text. To maintain the completeness and integrity of the overall pipeline, we retain the captioning process.

---

**Algorithm 2** Aug-PE with Image Voting (`Aug_PE_Image_Voting`)

---

**Require:** Private image dataset $\mathcal{D}$, generated private text $\mathcal{T}$, population size $N$, number of iterations $G$, `RANDOM_API`$(\cdot)$, `VARIATION_API`$(\cdot)$, diffusion model API `Diffusion`$(\cdot)$, image encoder `EncodeImage`$(\cdot)$, DP noise multiplier $\sigma, K, L$

**Ensure:** Differentially private synthetic text data $\mathcal{T}'$

1: $\mathcal{C}_0 = $ `RANDOM_API`$(N)$
2: $\mathcal{E}_{\text{priv}} = $ `EncodeImage`$(\mathcal{D})$
3: **for** $g = 0$ to $G - 1$ **do**
4:     **if** $K == 0$ **then**
5:         $\mathcal{I}_g = $ `Diffusion`$(\mathcal{C}_g)$
6:         $\mathcal{E}_{\text{gen}} = $ `EncodeImage`$(\mathcal{I}_g)$
7:     **else if** $K > 0$ **then**
8:         $\mathcal{C}_g^k = $ `VARIATION_API`$(\mathcal{C}_g), k = 1, 2, ..., K$
9:         $\mathcal{I}_g^k = $ `Diffusion`$(\mathcal{C}_g^k), k = 1, 2, ..., K$
10:        $\mathcal{E}_{\text{gen}}^k = $ `EncodeImage`$(\mathcal{I}_g^k), k = 1, 2, ..., K$
11:        $\mathcal{E}_{\text{gen}} = \frac{1}{K} \sum_{k=1}^{K} \mathcal{E}_{gen}^k$
12:     **end if**
13:     $H = \mathbf{0}^N = [0, 0, ..., 0]$
14:     **for** each $e_{priv} \in \mathcal{E}_{\text{priv}}$ **do**
15:         $i \leftarrow \arg\min_j d(e_{priv}, \mathcal{E}_{\text{gen}}[j])$
16:         $H[i] \leftarrow H[i] + 1$
17:     **end for**
18:     $H \leftarrow H + \mathcal{N}(0, \sigma^2 \mathbf{I})$
19:     $P \leftarrow H / \sum H$
20:     **if** $L == 1$ **then**
21:         $\mathcal{C}_g' \leftarrow$ draw $N$ samples with replacement from $\mathcal{C}_g$
22:         $\mathcal{C}_{g+1} \leftarrow$ `VARIATION_API`$(\mathcal{C}_g')$
23:         save $\mathcal{C}_{g+1}^{syn} \leftarrow \mathcal{C}_g'$
24:     **else if** $L > 1$ **then**
25:         $\mathcal{C}_g' \leftarrow$ **rank samples** by probabilities $P$ and draw top $N$ samples
26:         $\mathcal{C}_{g+1}^l \leftarrow$ `VARIATION_API`$(\mathcal{C}_g'), l = 1, 2, ..., L - 1$
27:         $\mathcal{C}_{g+1} = [C_g', C_{g+1}^1, ..., C_{g+1}^{L-1}]$
28:         save $\mathcal{C}_{g+1}^{syn} \leftarrow \mathcal{C}_{g+1}$
29:     **end if**
30: **end for**
31: **return** Final text candidates $\mathcal{C}_G$ as synthetic data $\mathcal{T}'$

---

# B  More on our experiment results

Here we post more results of our experiments, and provide more experiment details.

## B.1  FID comparison between PE and SPTI

Here we present details of our experiment results. We use FID [16] to compare between *SPTI* and PE on a variaty of datasets. We give specific dataset name and DP constraint in the caption of each image. We find *SPTI* generally better than PE, and voted samples is generally better than variants in most iterations. Interestingly, voted samples is often slightly worse than variants in the inital iteration.

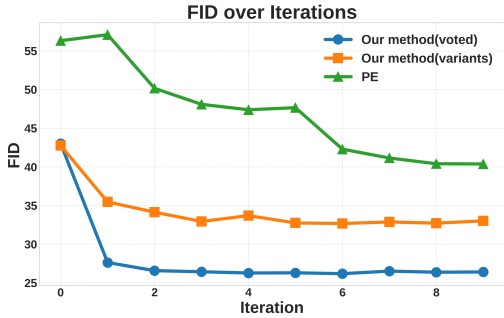

Figure 8: Experiment results on **LSUN Bedroom** dataset. ($\epsilon = 1.0$)

Figure 9: Experiment results on **LSUN Bedroom** dataset. ($\epsilon = 10.0$)

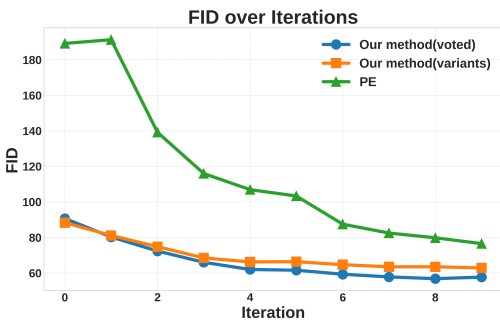

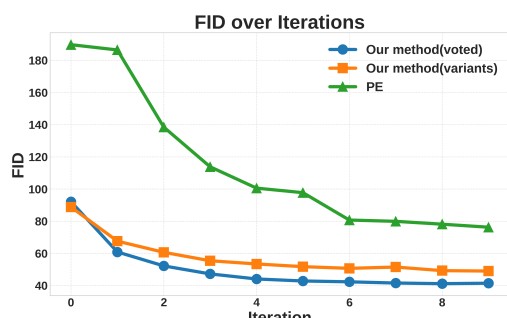

Figure 10: Experiment results on **European Art** dataset. ($\epsilon = 1.0$)

Figure 11: Experiment results on **European Art** dataset. ($\epsilon = 10.0$)

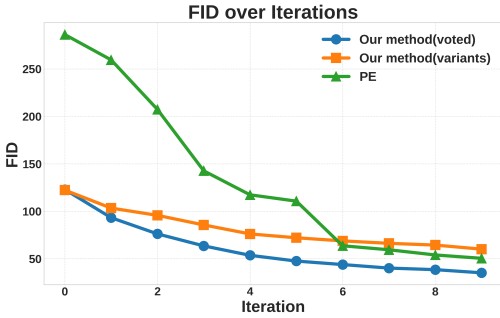

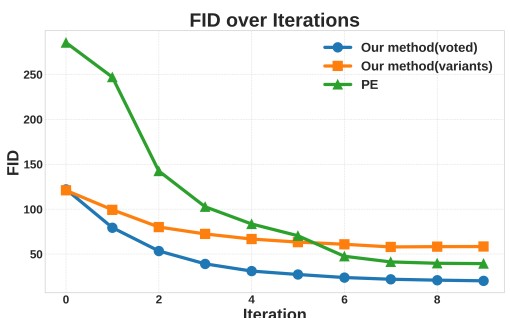

Figure 12: Experiment results on **Wave-ui-25k** dataset. ($\epsilon = 1.0$)

Figure 13: Experiment results on **Wave-ui-25k** dataset. ($\epsilon = 10.0$)

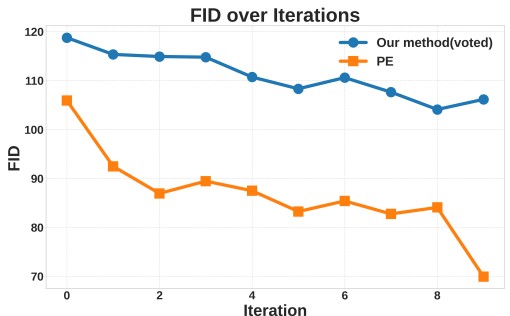

Figure 14: Experiment results on **Cat** dataset. ($\epsilon = 1.0$)

Figure 15: Experiment results on **Cat** dataset. ($\epsilon = 10.0$)

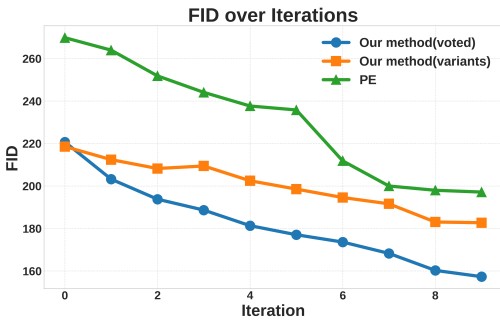
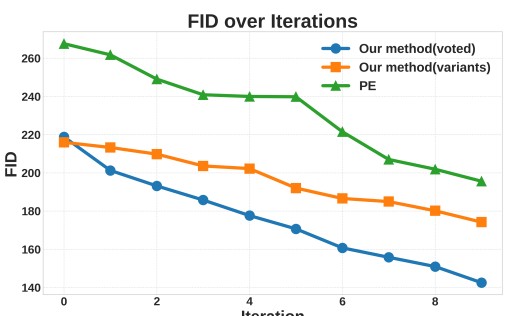

Figure 16: Experiment results on **Sprite Fright** dataset. ($\epsilon = 1.0$)

Figure 17: Experiment results on **Sprite Fright** dataset. ($\epsilon = 10.0$)

## B.2 FID comparison between Image Voting and Text Voting

Here we present details of our experiment results. We use FID [16] to compare between image voting and text voting on a variaty of datasets. We give specific dataset name and DP constraint in the caption of each image. We find image voting is generally better than text voting under most conditions. Interestingly, image voting and text voting show the same behavior in the inital iteration, i.e. voted samples is slightly worse than variants in the first iteration, no matter image voting or text voting is used.

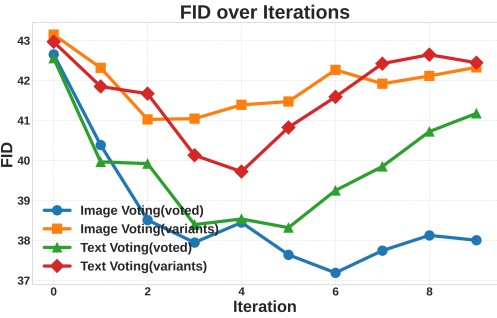
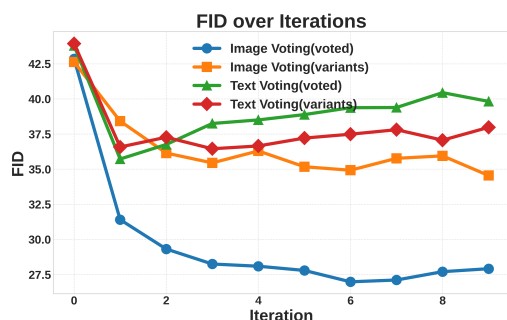

Figure 18: Experiment results on **LSUN Bedroom** dataset. ($\epsilon = 1.0$)

Figure 19: Experiment results on **LSUN Bedroom** dataset. ($\epsilon = 10.0$)

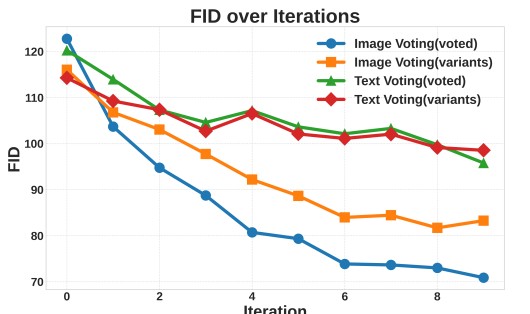

Figure 20: Experiment results on **Wave-ui-25k** dataset. ($\epsilon = 1.0$)

Figure 21: Experiment results on **Wave-ui-25k** dataset. ($\epsilon = 10.0$)

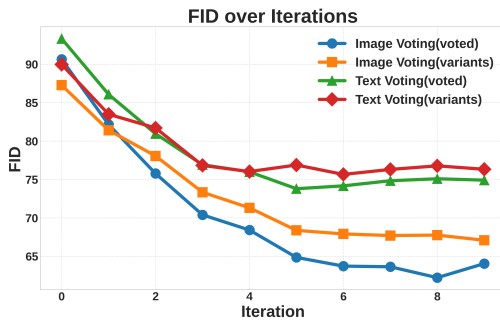

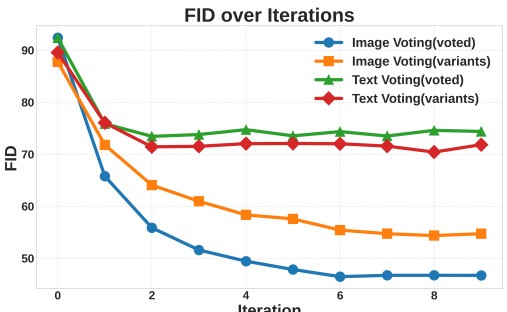

Figure 22: Experiment results on **European Art** dataset. ($\epsilon = 1.0$)

Figure 23: Experiment results on **European Art** dataset. ($\epsilon = 10.0$)

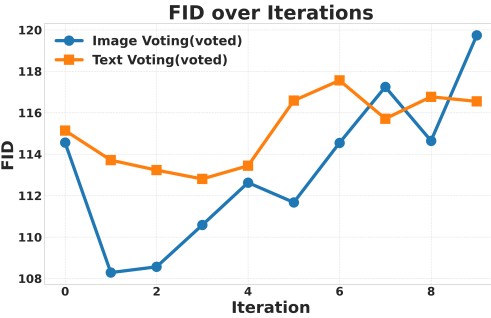

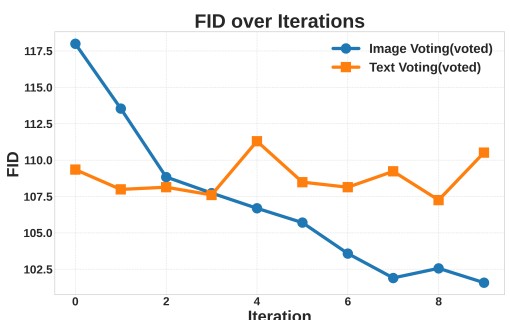

Figure 24: Experiment results on **Cat** dataset. ($\epsilon = 1.0$)

Figure 25: Experiment results on **Cat** dataset. ($\epsilon = 10.0$)

## B.3 Downstream Task

Despite the high quality of generated images in our method, we also consolidate their value in usage for downstream tasks. To be more specific, we test the classification accuracy on **CelebA** dataset using **WRN-40-4** model. Here we provide figures of model performance during training.

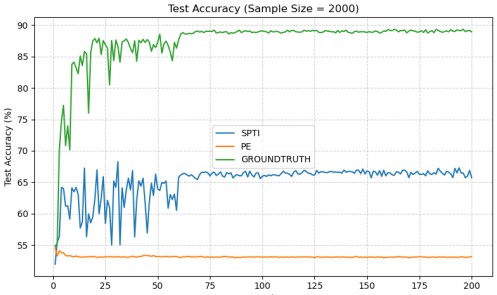

Figure 26: Training performance (test accuracy) on **CelebA** dataset with the same number of generated samples. ($\epsilon = 10.0$, `sample_size=2000`)

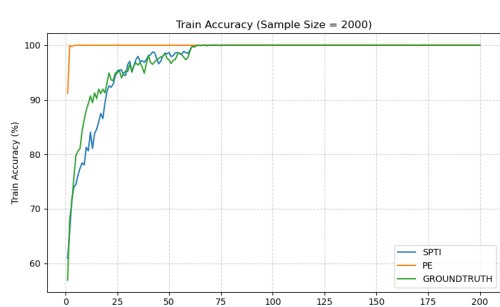

Figure 27: Training performance (train accuracy) on **CelebA** dataset with the same number of generated samples. ($\epsilon = 10.0$, `sample_size=2000`)

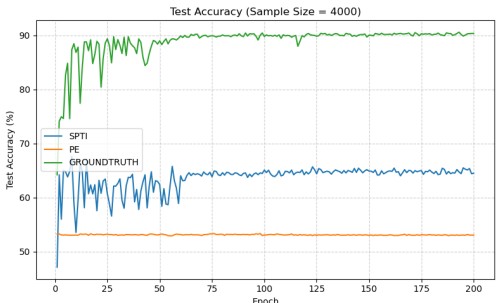

Figure 28: Training performance (test accuracy) on **CelebA** dataset with the same number of generated samples. ($\epsilon = 10.0$, `sample_size=4000`)

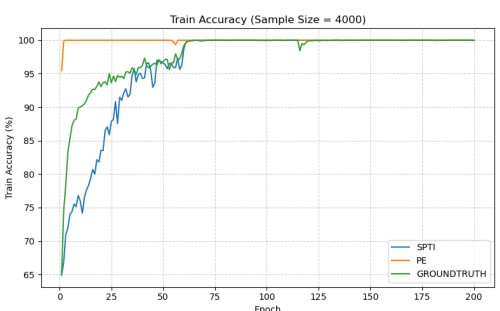

Figure 29: Training performance (train accuracy) on **CelebA** dataset with the same number of generated samples. ($\epsilon = 10.0$, `sample_size=4000`)

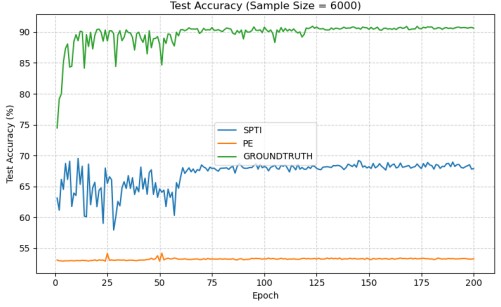

Figure 30: Training performance (test accuracy) on **CelebA** dataset with the same number of generated samples. ($\epsilon = 10.0$, `sample_size=6000`)

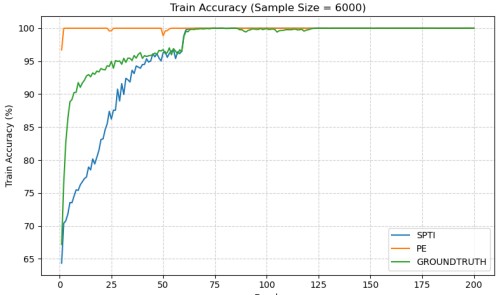

Figure 31: Training performance (train accuracy) on **CelebA** dataset with the same number of generated samples. ($\epsilon = 10.0$, `sample_size=6000`)

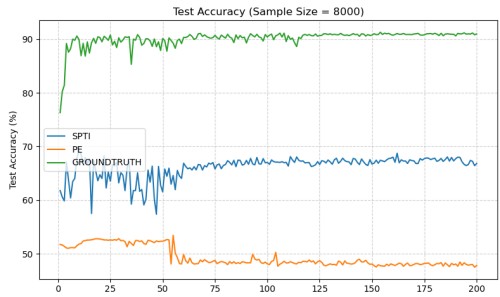

Figure 32: Training performance (test accuracy) on **CelebA** dataset with the same number of generated samples. ($\epsilon = 10.0$, `sample_size=8000`)

Figure 33: Training performance (train accuracy) on **CelebA** dataset with the same number of generated samples. ($\epsilon = 10.0$, `sample_size=8000`)

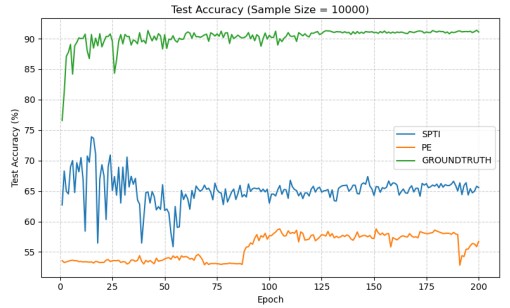

Figure 34: Training performance (test accuracy) on **CelebA** dataset with the same number of generated samples. ($\epsilon = 10.0$, `sample_size=10000`)

Figure 35: Training performance (train accuracy) on **CelebA** dataset with the same number of generated samples. ($\epsilon = 10.0$, `sample_size=10000`)

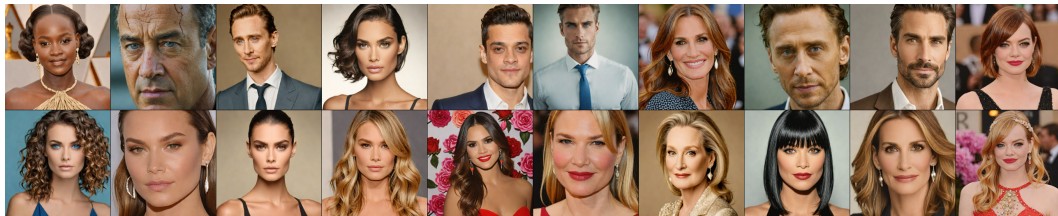

Figure 36: Generated samples by *SPTI* ($\epsilon = 10.0$, `sample_size=10000`)

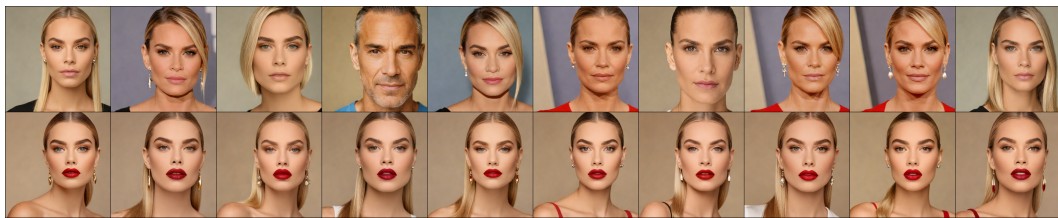

Figure 37: Generated samples by PE image ($\epsilon = 10.0$, `sample_size=10000`)

# C Extended Ablation Study

## C.1 Large Language Model and Diffusion Model APIs

Along with the effect of our image voting trick, we also tested our method with different LLM and diffusion model APIs along with other hyperparameters. For large language models, we choose **qwen-plus** [3] as comparison to our original **Meta-Llama-3-8B-Instruct** [2] model. For diffusion model APIs, we choose **stable-diffusion-xl-base-1.0** [44] to compare with our original model. We also present Figure 38, 39, 40, 41 generated from DP synthetic data by Imagen3 [19] and DALLE3 [37], i.e. we acquire DP synthetic images by running *SPTI* using **Meta-Llama-3-8B-Instruct** [2] and **stable-diffusion-xl-base-1.0** [44], but generate final DP synthetic images using different image-generation API.

|  | SDXL-Turbo | SDXL-base-1.0 | Infinity |
|---|---|---|---|
| **Meta-Llama-3-8B-Instruct** | 26.71 | 25.42 | 30.66 |
| **qwen-plus** | 26.65 | 24.44 | 31.28 |

Table 4: **FID** (lower is better) on **LSUN Bedroom** with $\epsilon = 1.0$, tested using different LLM and diffusion APIs. All other settings are kept identical to the default configuration. The results indicate that while different API backends (e.g., SDXL-Turbo, SDXL-base-1.0, Infinity) and language models (e.g., Meta-Llama-3-8B-Instruct, Qwen-Plus) introduce minor variations in FID, their overall influence on the final image quality is limited. This demonstrates the robustness of our method to the choice of model and API implementation.

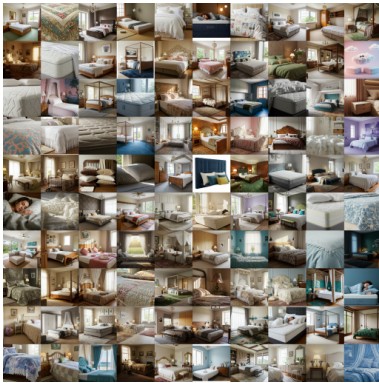

Figure 38: Images generated by Imagen3 [19] from DP-synthetic text of dataset **LSUN Bedroom** ($\epsilon = 1.0$). **FID** (lower is better) is 26.58, which is slightly better than using SDXL-Turbo 26.71.

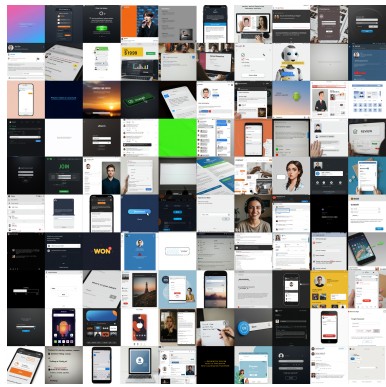

Figure 39: Images generated by Imagen3 [19] from DP-synthetic text of dataset **Wave-ui-25k** ($\epsilon = 1.0$). **FID** (lower is better) is 72.13, which is worse than using SDXL-Turbo 36.52.

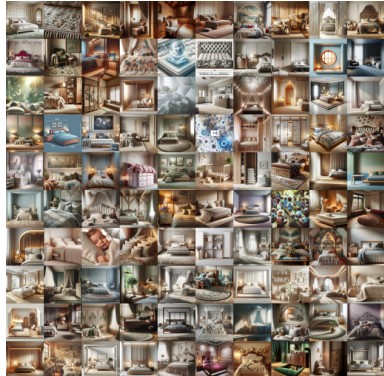 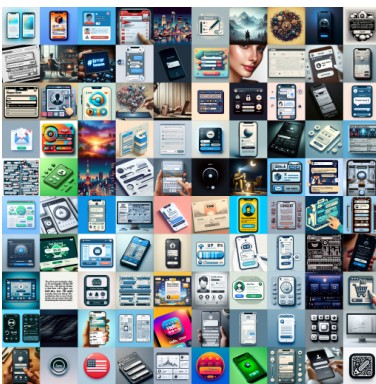

Figure 40: Images generated by DALLE3 [37] from DP-synthetic text of dataset **LSUN Bedroom** ($\epsilon = 1.0$). FID (lower is better) is 29.96, which is slightly worse than using SDXL-Turbo 26.71

Figure 41: Images generated by DALLE3 [37] from DP-synthetic text of dataset **Wave-ui-25k** ($\epsilon = 1.0$). FID (lower is better) is 150.68, which is worse than SDXL-Turbo 36.52

## D Explanation of Special Conditions

In the previous experiment on **Cat** dataset, we find our method to be less effective compared to baseline method (**PE**). So we conducted a contrast experiment to analyze the cause behind the phenomenon.

Firstly, we build a small sub-dataset from **ImageNet100** dataset called **bird** dataset. This dataset consists of 100 images labeled *goldfinch* and 100 images labeled *indigo bird*, which corresponds to label id 70 and 2 in the **ImageNet100** dataset. We found similar results to **Cat** dataset, so we take **bird** dataset as an alternate baseline.

Secondly, we proposed two possible reasons for the phenomenon that *SPTI* performs poorer than **PE** on **Cat** and **bird** dataset. One is that the composition of dataset is not diverse, while *SPTI* uses Aug-PE to generate DP data, which brings more diversity than PE image method, causing worse results. The other is that the *SPTI* performs naturally poorer than PE when the total number of dataset is small. We build another 3 datasets 5 according to these two reasons.

Here we reference our newly-built dataset using their composition in the following table. We only use images from **ImageNet100** when building all the four dataset. The experiment settings follows the experiments on **Cat** dataset in Table 6. The experiment results in Table 5 shows that the reason why *SPTI* performs poorer than PE on **Cat** and **bird** dataset is because diversity. *SPTI* performs naturally poorer on dataset that is less diverse. This provides us with insight that text modality may bring more diversity on DP data generation than using image.

## E Details of Experimental Configurations

To facilitate reproducibility, we detail the hyperparameter settings used for all experiments in Section 4. For hyperparameters not mentioned, the default values are used. For experiment settings of ablation studies between image voting and text voting, we use the same hyperparameters in Table 6 below. For comparison experiments on *SPTI*, PE and DP fine-tuning, we present our configuration details in Table 7. For specific configuration of DP fine-tuning, we present our settings in Table 8.

## F Datasets

In this section, we summarize datasets used in our experiments with their stats, i.e. total number of images, resolution, etc. We represent all information in Table 9. Here we record the number of images and resolution in our experiments.

Table 5: **FID** (lower is better) comparison of SPTI and PE on Different Datasets

| Dataset Composition | Total Number of Images | SPTI (ours) | PE |
|---|---|---|---|
| goldfinch*100 + indigo bird*100 | 200 | 144.18 | **110.96** |
| 20 kinds of different birds, each kind 10 images | 200 | **111.84** | 168.39 |
| goldfinch*1000 + indigo bird*1000 | 2000 | 106.26 | **78.66** |
| 20 kinds of different birds, each kind 100 images | 2000 | **84.25** | 172.93 |

| Dataset | Configurations | | *SPTI*(ours) | PE image |
|---|---|---|---|---|
| LSUN bedroom | PE.run() | num_samples_schedule | 2000 | 2000 |
| | | iterations | 10 | 10 |
| | ImageVotingNN() | lookahead_degree | 0 | - |
| | NNhistogram() | lookahead_degree | - | 4 |
| | PEPopulation() | initial_variation_api_fold | 6 | 0 |
| | | next_variation_api_fold | 6 | 1 |
| Cat | PE.run() | num_samples_schedule | 200 | 200 |
| | | iterations | 10 | 10 |
| | ImageVotingNN() | lookahead_degree | 4 | - |
| | NearestNeighbors() | lookahead_degree | - | 4 |
| | PEPopulation() | initial_variation_api_fold | 0 | 0 |
| | | next_variation_api_fold | 1 | 1 |
| Wave-ui-25k | PE.run() | num_samples_schedule | 2000 | 2000 |
| | | iterations | 10 | 10 |
| | ImageVotingNN() | lookahead_degree | 0 | - |
| | NNhistogram() | lookahead_degree | - | 4 |
| | PEPopulation() | initial_variation_api_fold | 6 | 0 |
| | | next_variation_api_fold | 6 | 1 |
| Europeart | PE.run() | num_samples_schedule | 2000 | 2000 |
| | | iterations | 10 | 10 |
| | ImageVotingNN() | lookahead_degree | 0 | - |
| | NNhistogram() | lookahead_degree | - | 4 |
| | PEPopulation() | initial_variation_api_fold | 6 | 0 |
| | | next_variation_api_fold | 6 | 1 |
| Sprite Fright | PE.run() | num_samples_schedule | 1000 | 1000 |
| | | iterations | 10 | 10 |
| | ImageVotingNN() | lookahead_degree | 0 | - |
| | NNhistogram() | lookahead_degree | - | 4 |
| | PEPopulation() | initial_variation_api_fold | 6 | 0 |
| | | next_variation_api_fold | 6 | 1 |

Table 6: Experiment Settings for **FID** evaluation.

| Configurations | | *SPTI*(ours) | PE image |
|---|---|---|---|
| PE.run() | num_samples_schedule | 2000 | 2000 |
| | iterations | 10 | 10 |
| ImageVotingNN() | lookahead_degree | 0 | - |
| NNhistogram() | lookahead_degree | - | 4 |
| PEPopulation() | initial_variation_api_fold | 6 | 0 |
| | next_variation_api_fold | 6 | 1 |

Table 7: Experiment Settings of *SPTI* and PE for **FID** evaluation of **MM-Celeba-HQ** dataset

Table 8: Hyperparameters for fine-tuning diffusion models with DP constraints $\epsilon = 10, 1$ and $\delta = 10^{-5}$ on text-conditioned CelebAHQ.

|  | $\epsilon = 10$ | $\epsilon = 1$ |
|---|---|---|
| batch size | 256 | 256 |
| base learning rate | $1 \times 10^{-7}$ | $1 \times 10^{-7}$ |
| learning rate | $2.6 \times 10^{-5}$ | $2.6 \times 10^{-5}$ |
| epochs | 10 | 10 |
| clipping norm | 0.01 | 0.01 |
| noise scale | 0.55 | 1.46 |
| ablation | -1 | -1 |
| num of params | 280M | 280M |
| use_spatial_transformer | True | True |
| cond_stage_key | caption | caption |
| context_dim | 1280 | 1280 |
| conditioning_key | crossattn | crossattn |
| transformer depth | 1 | 1 |

| Dataset | Total number of images | Resolution | Source |
|---|---|---|---|
| **LSUN** | 300,000 | $256 \times 256$ | https://github.com/fyu/lsun |
| **Cat** | 200 | $512 \times 512$ | https://www.kaggle.com/datasets/fjxmlzn/cat-cookie-doudou/ |
| **Wave-ui-25k** | 24,798 | $256 \times 256$ | https://huggingface.co/datasets/agentsea/wave-ui-25k |
| **European Art** | 15,154 | $256 \times 256$ | https://huggingface.co/datasets/biglam/european_art |
| **Sprite Fright** | 13,077 | $256 \times 256$ | https://studio.blender.org/projects/sprite-fright/ |
| **MM-Celeba-HQ** | 30,000 | $256 \times 256$ | https://github.com/IIGROUP/MM-CelebA-HQ-Dataset |
| **CelebA** | 202,599 | $178 \times 178$ | https://mmlab.ie.cuhk.edu.hk/projects/CelebA.html |

Table 9: Statistics of datasets used in our experiments.

