# OpenReview forum: "Synthesize Privacy-Preserving High-Resolution Images via Private Textual Intermediaries"
_NeurIPS.cc/2025/Conference — NeurIPS 2025 poster_

### Official Review · Reviewer_ocKp · 2025-06-13

**Clarity:** 3
**Significance:** 2
**Originality:** 3
**Rating:** 4
**Confidence:** 4

**Summary:**

This paper introduces SPTI (Synthesis via Private Textual Intermediaries), a novel framework for privacy-preserving differentially private (DP) image synthesis. The method first captions private images using a vision-language model (VLM), then applies an Augmented Private Evolution algorithm to generate DP captions, and finally generates synthetic images from the DP text via a diffusion model. Notably, the entire pipeline is training-free and capable of producing high-resolution outputs. Experiments show favorable performance based on FID scores. The authors also contribute a new dataset, Sprite Fright, for benchmarking.

**Questions:**

1. Why are no downstream tasks included to assess the semantic utility of the synthesized images? (Related to Major Weakness 1)

2. Given the known generalization limitations of pretrained models, can you provide examples or analysis of failure cases—i.e., images whose captions omit critical visual information? (Also related to Major Weakness 1)

3. Why are the DP-finetuning results mentioned in the text but not included in Table 1? (See Major Weakness 2)

4. Why was Euclidean distance selected for similarity search in embedding space, instead of cosine similarity? (See Major Weakness 3)

5. The FID metric has known shortcomings [1]. Did you consider alternative or improved evaluation metrics (e.g., CMMD [1])?

6. The substantial performance gap between SPTI and the PE image on the Cat dataset (Table 1) warrants further explanation.

[1] Jayasumana, S., Ramalingam, S., Veit, A., Glasner, D., Chakrabarti, A., & Kumar, S. (2024). Rethinking fid: Towards a better evaluation metric for image generation. In Proceedings of the IEEE/CVF Conference on Computer Vision and Pattern Recognition (pp. 9307-9315).

**Ethical Concerns:**

["NO or VERY MINOR ethics concerns only"]

**Final Justification:**

All my concerns have been addressed in the rebuttal. Furthermore, the discussion between the authors and other reviewers reinforces the strength of the proposed image voting mechanism. I have therefore raised my score to 4.

**Limitations:**

The authors appropriately acknowledge that the captioning process may not preserve all image information and that domain-specific details could be lost. However, this limitation could be better quantified through targeted experiments, especially using downstream tasks that rely on high-fidelity semantics.

**Quality:**

3

**Strengths And Weaknesses:**

Strengths:
1. The idea of using text as an intermediary for privacy-preserving image synthesis is both novel and practical, particularly because it avoids the need for model training and leverages powerful pretrained models.

2. The proposed Aug-PE algorithm for DP text generation is a meaningful technical contribution.

3. The authors contribute a new dataset, Sprite Fright, which could benefit future work in DP image synthesis.

Weaknesses:
Majors:
1. Information loss via captioning is underexplored.
Using VLMs to extract captions from images can introduce significant information loss, especially in specialized domains or when fine-grained details are involved. While this is briefly acknowledged in the limitations section, it deserves deeper empirical investigation. The current evaluation relies solely on the FID metric, which does not capture semantic fidelity. Including downstream tasks (e.g., classification) would better evaluate the utility of the generated images.

2. Missing results for DP-finetuning.
The experiments section mentions a comparison against DP-finetune, but Table 1 does not report these results. This inconsistency should be addressed.

3. Choice of similarity metric.
Section 3.2 states that Euclidean distance is used for nearest-neighbor search based on embeddings. However, cosine similarity is more commonly used in text/image embedding spaces. Justifying this choice—or comparing both—would strengthen the methodology.

Minors:
1. The last paragraph of the Introduction lacks a summary of Section 5. Adding this would improve flow and completeness.
2. Inconsistent naming of prior work—please standardize references to "PE" or "PE image" throughout the paper.

---

> ### Author Rebuttal · Authors · 2025-07-26
>
> We appreciate the reviewer’s insightful feedback. Below, we provide detailed responses to each of the raised points.
>
> **Response to Weakness 1: more verification of information loss via captioning.**
>
> Thank you for the question. We have conducted additional experiments on downstream tasks and evaluated the proposed CMMD metric to further validate that our approach preserves semantic fidelity. The results are presented below in Question 5.
>
> **Response to Weakness 2: Missing results for DP-finetuning.**
>
> We provide evaluation of DP-finetuning below in response to Question 3.
>
> **Response to Weakness 3: Choice of similarity metric.**
>
> We provide our argument below in response to Question 4.
>
> ## Question 1
> We're sorry for not implementing downstream tasks, we will provide more systematic experiments in the future. We prepare to use CelebA dataset and conduct a series of 2-class classification tasks according to its features. We're currently running experiments on `Wearing_Lipstick` feature, and acquired good results. We will update this series of experiments in the future version of our paper.
>
> ## Question 2
> We test our method on various datasets to confirm that our pipeline works well in different circumstances, and experiments did show good results. However, we admit that there are cases where text description doesn't align well with the images. This usually happens when images have such intensive information that VLMs can hardly capture all of them, or limits on text length prohibits VLMs to fully describe all the scenes. Nevertheless, we believe the performance of our method will improve as state-of-the-art VLMs and diffusion models reach better results.
> ## Question 3
> Our method focuses on conditions where private data may not be directly accessed. In our method, we refer to private data through it representations. In contrast, DP-finetune has direct access to private data, which can make full use of it. This is the key reason why we focus on comparison between SPTI and PE image, since our method is an improvement on PE image method.
>
> Nevertheless, SPTI performs better than DP-finetune in many circumstances, and we also conducted more experiments after paper submission, and we provide experiment results below.
>
> ||SPTI|PE Image| DP-finetune|
> |:-:|:-:|:-:|:-:|
> |LSUN Bedroom| **26.39** | 40.36 | 31.76 |
> |Wave-ui-25k| **37.39** | 50.45 | 62.08 |
> |European Art| **57.64** | 76.50 | 63.97 |
> |MM-Celeba-HQ| **34.17** | 50.45 | 57.01 |
>
> ## Question 4
> We adopt the same similarity metric used in the Private Evolution (PE) work. Since this component is not the focus of our technical contribution, we retain it unchanged to ensure a fair and consistent comparison with PE.
>
> ## Question 5
> We're sorry for not considering more evaluation metrics before, so we added more evaluation and re-run some of our previous experiments for results. Here we provide experiment results with both FID and CMMD evaluation. The consistency in these two metrics show that our method indeed improves PE image method.
>
> The table below shows CMMD value for SPTI and PE method. In this table, we found our method performs the best except for the case in LSUN Bedroom. We suppose it's because LSUN Bedroom has such a large quantity of image data that improves the performance of fine-tuning, but this needs further experiments. Other conditions show the excellence of our method.
>
> | Dataset | SPTI | PE | DP-finetune |
> |:---:|:---:|:---:|:---:|
> | LSUN Bedroom | 1.744 | 2.066 | **1.376** |
> | Wave-ui | **1.480** | 3.132 | 2.149 |
> | European_art | **0.684** | 1.016 | 2.447 |
> ## Question 6
> We propose that PE image works better than SPTI when the diversity of the private data is too low. For example, SPTI performs poorer than PE image because Cat dataset only involves images of 2 kinds of cat, Maine Coon and Siamese Cat. To validate this idea, we conducted a simple experiment. We build four datasets using images from **ImageNet100** dataset, and use exactly the same setting to run our experiment (with $\epsilon=1.0$, `num_samples=200`, same text prompt and FID metric). We provide its name, data composition and results below.
>
> | Dataset Name | Composition | Total Num | SPTI | PE Image |
> |:--:|:--:|:--:|:--:|:--:|
> | TwoSpecies_Bird_200 | 100\*goldfinch image+100\*indigo bird image| 200 | 144.18 | **110.96** |
> |TwentySpecies_Bird_200| 20 kinds of birds, each 10 images| 200| **111.84** | 168.39 |
> |TwoSpecies_Bird_2K| 1000\*goldfinch image+1000\*indigo bird image|2000| 106.26 | **78.66** |
> |TwentySpecies_Bird_2K| 20 kinds of birds, each 100 images | 2000| **84.25** | 172.93 |
>
> `TwoSpecies_Bird_200` dataset shows the same results as Cat dataset, so `TwoSpecies_Bird_200` is capable to be an alternative of Cat dataset. In this table, PE image performs better than SPTI when there's **only 2 kinds of bird**, no matter how many images are there. In contrast, SPTI performs better than PE image when there are **more kinds of bird**, no matter the total number of images. This validate our hypothesis, SPTI is suitable for diverse data while PE image suits for image that are less diverse.

---

> > ### Comment · Reviewer_ocKp · 2025-08-04
> > **Official Comment by Reviewer ocKp**
> >
> > Thank the authors for their rebuttal. Most of my concerns have been addressed. The experiment on the diversity issue is quite promising and makes SPTI an attractive method. However, the effectiveness of SPTI is heavily dependent on the captioning capability of the underlying VLM. I acknowledge that future VLMs may be able to capture image details more comprehensively. Given this situation, is there a way to assess how detailed a VLM can be in practice? Such an assessment would allow potential users to quickly determine whether their datasets are suitable for processing with SPTI.

---

> > > ### Author Response · Authors · 2025-08-05
> > > **Validating Reviewer's Insight: Image Voting Reduces Dependence on Captioning**
> > >
> > > We appreciate the reviewer’s insightful observation regarding the potential limitations of text voting due to the captioning capability of the underlying VLM. We agree that text voting can be significantly affected by imperfect captions. However, in the case of image voting, our method does **not rely on any text captioning model** (as can be verified in Section 3.2 and Appendix A). As shown in our experiments, image voting consistently outperforms text voting, and we believe this performance gap is largely due to the reason pointed out by the reviewer. This not only supports the reviewer’s intuition but also demonstrates that our image voting approach effectively mitigates the captioning dependency concern.

---

> > > > ### Comment · Reviewer_ocKp · 2025-08-05
> > > > **Official Comment by Reviewer ocKp**
> > > >
> > > > Thank the authors for the detailed response. I appreciate the design of image voting process and have no further concern.

---

### Official Review · Reviewer_dX13 · 2025-06-25

**Clarity:** 4
**Significance:** 3
**Originality:** 3
**Rating:** 4
**Confidence:** 4

**Summary:**

This paper proposes an interesting new idea to address the problem of differentially private synthetic image generation from text captions. Generating high quality DP images is hard since this leads to a poor privacy-utility tradeoff; on the other hand, DP text generation is relatively easy. This paper proposes a really nice idea of reducing private image generation to text-generation -- by first using a pre-trained model to caption an image, then a private image, then using a DP-finetuned model to produce a synthetic differentially private caption, and finally to use this caption and a pretrained text to image model to generate a similar synthetic image.

The main idea behind the paper is quite nice and the empirical results are quite strong.

**Questions:**

n/a

**Ethical Concerns:**

["NO or VERY MINOR ethics concerns only"]

**Final Justification:**

This is a solid paper and I am supportive of its acceptance.

**Limitations:**

One of the limitations of the paper is what happens when one of the pre-trained models are out-of-distribution wrt the private data. this could be discussed in more detail.

**Quality:**

3

**Strengths And Weaknesses:**

Strengths:

+ Elegant idea
+ Strong empirical results that beats existing image baselines
+ Well-written and clear presentation

Weaknesses (mostly minor):

- I know it is a bit difficult to fairly compare with image-only baselines, but some discussion on this would improve the paper further
- The related work is missing references to other image-only methods, such as [1, 2, 3]
[1] F. Harder, M. J. Asadabadi, D. J. Sutherland, and M. Park. Differentially private data generation
needs better features. arXiv preprint arXiv:2205.12900, 2022
[2] https://arxiv.org/pdf/2309.00008
[3] ] T. Cao, A. Bie, A. Vahdat, S. Fidler, and K. Kreis. Don’t generate me: Training differentially private generative models with sinkhorn divergence. Advances in Neural Information Processing Systems, 34:12480–12492, 2021

---

> ### Author Rebuttal · Authors · 2025-07-26
>
> ## Response to Comment: Fair comparison with image-only baselines and missing related work on image-only methods
>
> Thank you very much for pointing out these related literature. We appreciate the reviewer’s thoughtful comment on fairly comparing against image-only baselines, especially the references [1,2,3]. These methods showcase how differentially private generative models can be trained to produce synthetic image datasets le. Specifically:
>
> * \[1] and \[2] advocate for using public datasets or pre-trained public feature extractors to enhance DP training and achieve improved image quality.
> * \[3] introduces Sinkhorn divergence as a robust objective for training DP generative models, with promising results on image-only datasets.
>
> Our work shares a similar high-level motivation—**leveraging public or existing models to improve private data generation**—but advances the field in several notable ways:
>
> 1. **Methodologically**: Our framework can directly utilize state-of-the-art foundation models via API access, without requiring any training or fine-tuning.
> 2. **Conceptually**: We bridge modalities by introducing a **text-based intermediary representation**, which allows us to exploit the strong alignment between vision and language models. This cross-modal design expands the range of generative capabilities and circumvents limitations of purely image-based pipelines.
> 3. **Empirically**: To the best of our knowledge, our method is the first to demonstrate the ability to generate **high-resolution image datasets with differential privacy guarantees**, going beyond the typical CIFAR-10 resolution benchmark.
>
> We will integrate this comparison and discussion into the related work section to clarify how our approach builds upon and differs from these prior contributions.

---

### Official Review · Reviewer_Deqn · 2025-06-30

**Clarity:** 2
**Significance:** 2
**Originality:** 3
**Rating:** 4
**Confidence:** 4

**Summary:**

Private Evolution is a promising approach to generating synthetic image and text data under differential privacy. This paper proposes a method, SPTI, which combines image and text private evolution methods. This method 1. maps private images to text using an off-the-shelf mutli-modal model, 2. uses an novel version of private evolution to obtain synthetic text, then 3. maps the synthetic text to images. The primary technical contribution is a version of private evolution that uses image voting, where both text and images are utilized to successively generate synthetic samples.

**Questions:**

1. In Table 1, where are the results for DP-finetune?
2. Any thoughts on why Private Evolution works better on the Cat dataset than the proposed method?

Additional Comments
- Rather than stating privacy results in terms of Approximate-DP, I suggest stating results in terms of RDP and provide the conversion from RDP to Approximate-DP. Moreover, it would be helpful to have a formal statement of the privacy result and a proof in either the main text or an appendix.

**Ethical Concerns:**

["NO or VERY MINOR ethics concerns only"]

**Final Justification:**

The reviewers addressed my concerns about the privacy analysis in the their rebuttal.

**Limitations:**

yes

**Quality:**

2

**Strengths And Weaknesses:**

Strengths
- Interesting idea combining text and image approaches to private evolution.
- Figure 1 is clear and helpful for understanding the proposed method.
- The proposed method outperforms baseline methods with respect to FID score, and the image voting approach to private evolution outperforms using text alone.

Weaknesses
- There is not a formal proof in the main text or appendix that the proposed mechanism satisfies differential privacy. Moreover, the privacy analysis provided is insufficient. Most results such as the post-processing property are not clearly stated and are missing proper citations. Though stated on Line 188, the Advanced Composition Theorem (for Approx. DP) is not used for privacy accounting since the authors state that the privacy analysis uses RDP rather than Approx. DP.
- In the limitations section, the authors mention that the proposed method requires significantly more resources than baseline Private Evolution since the former requires several calls to muti-modal models; yet, there is no comparison of running time or token usage between the models.
- There are several grammar mistakes throughout the text.

---

> ### Author Rebuttal · Authors · 2025-07-26
>
> # To Reviewer
>
> We appreciate the reviewer’s insightful feedback. Below, we provide detailed responses to each of the raised points.
>
> ## Reply to Weaknesses
> ### Weakness 1
> **w1**: There is not a formal proof in the main text or appendix that the proposed mechanism satisfies differential privacy. Moreover, the privacy analysis provided is insufficient. Most results such as the post-processing property are not clearly stated and are missing proper citations. Though stated on Line 188, the Advanced Composition Theorem (for Approx. DP) is not used for privacy accounting since the authors state that the privacy analysis uses RDP rather than Approx. DP.
>
> **WA1**: We provide a detailed privacy analysis below. This analysis provides a tighter bound for DP guarantee, and is updated to the latest version of our paper.
>
> #### Privacy Analysis
>
> We analyze the privacy guarantees of our proposed *SPTI* framework, which synthesizes image data under differential privacy (DP) by operating in the text space. Our goal is to ensure that the final synthetic images $\mathcal{D}'$ are generated through a process that satisfies $(\varepsilon, \delta)$-DP with respect to the private dataset $\mathcal{D}$.
>
> **Privacy-Critical Step.** The only step in the *SPTI* pipeline that accesses the private data is the Private Evolution module, specifically through the similarity-based voting mechanism in algorithm `Aug_PE_Image_Voting`. This module operates on embeddings of the private image dataset $\mathcal{D}$ and is used to guide the sampling of synthetic text candidates. Thus, our privacy analysis focuses on this step.
>
> **Voting Mechanism.** For each private image embedding, we identify its nearest neighbor among the generated image embeddings , and increment a vote count histogram $H$ indexed by the candidate text that produced the nearest image. This step corresponds to a histogram query over the private data.
>
> To ensure differential privacy, we add Gaussian noise $\mathcal{N}(0, \sigma^2 \mathbf{I})$ to the vote histogram $H$ , which is equivalent to applying the \textit{Gaussian mechanism} to a function with bounded sensitivity.
>
> **Sensitivity Analysis.** The histogram query has $L_2$ sensitivity at most $1$, since each private image contributes a vote to only one candidate (i.e., changing a single image can affect the histogram by at most 1 in one coordinate). This satisfies the precondition for the Gaussian mechanism.
>
> According to the Gaussian mechanism \cite{balle2018improving}, adding noise of variance $\sigma^2$ per coordinate to a function with $L_2$ sensitivity $1$ ensures $(\varepsilon, \delta)$-DP, provided:
>
> $$
> \Phi\left(\frac{1}{2 \sigma}-{\varepsilon \sigma}\right)-e^{\varepsilon} \Phi\left(-\frac{1}{2 \sigma}-{\varepsilon \sigma}\right) \leq \delta .
> $$
>
> **Privacy Composition.** According to the adaptive composition theorem of Gaussian mechanisms \cite{dong2019gaussian}, applying the above Gaussian mechanism across $G$ iterations satisfy $(\varepsilon, \delta)$-DP, provided:
> $$
> \Phi\left(\frac{\sqrt{G}}{2 \sigma}-\frac{\varepsilon \sigma}{\sqrt{G}}\right)-e^{\varepsilon} \Phi\left(-\frac{\sqrt{G}}{2 \sigma}-\frac{\varepsilon \sigma}{\sqrt{G}}\right) \leq \delta .
> $$
>
> **Post-processing Immunity.** All downstream steps in *SPTI*—including text mutation and image generation via a fixed diffusion model—depend solely on the privatized output of the voting mechanism. By the *post-processing property* of DP, these steps incur no additional privacy cost.
>
> **Overall Guarantee.** Hence, **SPTI** satisfies $(\varepsilon, \delta)$-differential privacy with respect to the private image dataset $\mathcal{D}$, provided that the noise scale $\sigma$ is properly chosen to satisfy equation above.
>
> ### Weakness 2
>
> **w2**: In the limitations section, the authors mention that the proposed method requires significantly more resources than baseline Private Evolution since the former requires several calls to muti-modal models; yet, there is no comparison of running time or token usage between the models.
>
> **WA2**:
> Thank you for raising this point. We have conducted a systematic evaluation of the computational cost of our pipeline and will include the results in the next version of the paper.
>
> Our full data generation process involves two main components: an 8B-parameter large language model (LLM) and a 3.5B-parameter diffusion model. In practice, a complete generation run takes approximately 19 hours on a single NVIDIA A800, with a peak memory footprint of around 70 GB. In comparison, the Private Evolution baseline uses a 3.5B-parameter diffusion model, does not involve an LLM, and completes in about 7 hours on a single NVIDIA A800, with a peak memory usage of approximately 35 GB.
>
> While our method introduces a modest increase in computational cost—roughly doubling GPU memory and requiring ~2.7× runtime—this is a marginal trade-off given the qualitative generation gains. By leveraging multi-modal language-guided synthesis, our approach significantly improves semantic richness, data diversity, and downstream utility, justifying the additional resources.
>
>
> ### Weakness 3
>
> **w3**: There are several grammar mistakes throughout the text.
>
> **WA3**: We went through our paper again and fixed most grammar mistakes.
>
> ## Reply to Questions
> ### Question 1
>
>
> We ran more experiments on DP-finetune, and provide comparison between SPTI, PE Image and DP-finetune below. We fix the experiment settings as the same, with $\epsilon=1.0$, `num_samples=2000`. It's noticeable that we updated chat template for LLM, so we re-run all the previous experiments. Still, new results show that our method performs better than PE image and DP-finetune.
>
> ||SPTI|PE Image| DP-finetune|
> |:-:|:-:|:-:|:-:|
> |LSUN Bedroom| **26.39** | 40.36 | 31.76 |
> |Wave-ui-25k| **37.39** | 50.45 | 62.08 |
> |European Art| **57.64** | 76.50 | 63.97 |
> |MM-Celeba-HQ| **34.17** | 50.45 | 57.01 |
>
> ### Question 2
> PE image works better than SPTI when the diversity of the private data is too low. For example, SPTI performs poorer than PE image because Cat dataset only involves images of 2 kinds of cat, Maine Coon and Siamese Cat. To validate this idea, we conducted a simple experiment. We build four datasets using images from **ImageNet100** dataset, and use exactly the same setting to run our experiment (with $\epsilon=1.0$, `num_samples=200`, same text prompt and FID metric). We provide its name, data composition and results below.
>
> | Dataset Name | Composition | Total Num | SPTI | PE Image |
> |:--:|:--:|:--:|:--:|:--:|
> | TwoSpecies_Bird_200 | 100\*goldfinch image+100\*indigo bird image| 200 | 144.18 | **110.96** |
> |TwentySpecies_Bird_200| 20 kinds of birds, each 10 images| 200| **111.84** | 168.39 |
> |TwoSpecies_Bird_2K| 1000\*goldfinch image+1000\*indigo bird image|2000| 106.26 | **78.66** |
> |TwentySpecies_Bird_2K| 20 kinds of birds, each 100 images | 2000| **84.25** | 172.93 |
>
> `TwoSpecies_Bird_200` dataset shows the same results as Cat dataset, so `TwoSpecies_Bird_200` is capable to be an alternative of Cat dataset. In this table, PE image performs better than SPTI when there's **only 2 kinds of bird**, no matter how many images are there. In contrast, SPTI performs better than PE image when there are **more kinds of bird**, no matter the total number of images. This validate our hypothesis, SPTI is suitable for diverse data while PE image suits for image that are less diverse.

---

> > ### Comment · Reviewer_Deqn · 2025-08-08
> >
> > I thank the authors for their detailed rebuttal and for running additional experiments within the short review window.
> >
> > This rebuttal has addressed my concerns. I'll raise my score to a 4.

---

### Official Review · Reviewer_6vVq · 2025-07-02

**Clarity:** 3
**Significance:** 2
**Originality:** 2
**Rating:** 4
**Confidence:** 3

**Summary:**

This paper proposes synthesis via private textual intermediaries (SPTI) that is a framework for generating high-resolution differentially private synthetic images. The main idea is to shift the privacy-preserving synthesis process from the image domain to the text domain. The process begins by transforming private images into textual descriptions through an image captioning model. It then utilizes a modified Private Evolution algorithm to produce differential privacy (DP) text, followed by reconstructing images using a text-to-image diffusion model. Experiments on multiple datasets show that SPTI achieves better image quality (in terms of FID) than Private Evolution and DP fine-tuning.

**Questions:**

1) How sensitive is the overall performance to the choice of the image-to-text component? For example, if the captioning model were replaced with another model, such as LLaVA, how would this impact the image quality (FID)? Is there any quantitative analysis or ablation regarding this choice?

2) For datasets with image tags or class labels, what would happen if the text input were simply 11an image of [tag]'' or a similar template for each image? Would this significantly affect performance or privacy guarantees?

3) Are there any alternative evaluation methods, beyond FID scores, to assess the distribution alignment between the synthetic and private data? For example, can we use techniques like t-SNE or UMAP to visualize the embedding spaces of private and synthetic images in order to assess whether their distributions are well aligned? While the FID scores show improvements, it remains unclear whether the synthetic data truly captures the diversity and structure of the private data distribution.

4) What is the estimated budget, in terms of computation and cost, for processing a datasets used in the paper, specifically for the image-to-text conversion using APIs like GPT or other captioning models?

**Ethical Concerns:**

["Major Concern: Safety and security"]

**Final Justification:**

During the rebuttal, the authors addressed my concerns about the paper, which led me to raise my final rating.

**Limitations:**

No. The paper recognizes the significant computational costs and API token usage but fails to provide specific figures or estimates. Without this information, it is challenging to evaluate the method's practical expense and determine if the cost is a reasonable trade-off compared to other DP approaches.

**Paper Formatting Concerns:**

There are no obvious formatting issues.

**Quality:**

2

**Strengths And Weaknesses:**

**Strengths**
1) The paper introduces a new framework, Synthesis via Private Textual Intermediaries (SPTI), which transitions the challenge of differentially private image synthesis from images to text.
2) The method uses API-based models, including image captioning and text-to-image generation, without requiring model fine-tuning.
3) The pipeline is modular and does not require DP fine-tuning of large image generation models, which reduces technical complexity compared to DP-SGD-based approaches, such as DP Fine-tuning.
4) Experimental results show improvements over Private Evolution and DP fine-tuning, across multiple datasets and privacy budgets.

**Weaknesses**
1) The method has several modules: image-to-text, DP text evolution, and text-to-image synthesis. The interdependencies among these components make it challenging to assess their impact on overall performance.  In Section 4.3, the ablation study shows that the method is sensitive to changes in the image generation models.

2) Using image-to-text as an intermediate representation has been explored in some existing works [1]. Although the application scenarios differ, the underlying idea is similar.

      [1] Image2text2image: A novel framework for label-free evaluation of image-to-text generation with text-to-image diffusion models.

3) It is unclear whether text-based representations are always sufficient to preserve fine-grained image details.

4) The baseline comparisons are incomplete. While the primary comparisons focus on Private Evolution and DP fine-tuning, there is no evaluation against simpler baselines.
   * Sampling random text prompts from the LLM API (without using any private data) to generate images, or generating images solely based on class names, raises uncertainty about whether this basic approach can achieve competitive FID scores.

5) The DP guarantee is limited to the Private Evolution voting step, while other processes, particularly image-to-text captioning, access private data without DP protection. Although the paper cites the post-processing property of DP as justification, this design choice warrants further examination.

6) Computational costs are not fully discussed. While the method avoids DP training, it requires repeated calls to large captioning models, LLMs, and text-to-image diffusion models. The paper reports that experiments were run on high-end GPUs, but no systematic analysis of compute costs compared to prior methods is provided. There are also additional costs for calling APIs such as ChatGPT.

7) The paper does not fully analyze failure modes or limitations in the semantic fidelity of the generated images, especially when the text descriptions do not align well with the images in complex scene data.

---

> ### Author Rebuttal · Authors · 2025-07-26
>
> # To Reviewer
> We appreciate the reviewer’s insightful feedback. Below, we provide detailed responses to each of the raised points.
>
> ## Reply to Weaknesses
> ### Weakness 1
> **W1:** The interdependencies among these components make it challenging to assess their impact on overall performance. In Section 4.3, the ablation study shows that the method is sensitive to changes in the image generation models.
>
> **WA1:** This statement is inaccurate. In addition to analyzing the image generation models, we also conducted a sensitivity analysis on the image captioning component. Overall, the proposed method is not particularly sensitive to the choice of these base models, provided that each performs its respective task effectively.
>
> ### Weakness 2
>
> **W2:** Using image-to-text as an intermediate representation has been explored in some existing works [1]. Although the application scenarios differ, the underlying idea is similar.
>
>
> **WA2:** Thank you for referring to paper [1], which proposes a label-free evaluation framework for image captioning models. The core idea is that a well-written caption should enable a text-to-image diffusion model to reconstruct an image visually similar to the original.
>
> While the use of an image-to-text-to-image pipeline is conceptually related and increasingly common, its application to the task of private dataset synthesis is nontrivial and remains unclear. Our work differs substantially in terms of motivation, privacy guarantees, and evaluation methodology:
>
> Objective: Paper [1] aims to assess the quality of image captioning models. In contrast, our work is centered on synthesizing datasets that preserve data utility while offering rigorous differential privacy (DP) guarantees.
>
> Evaluation Metric: The prior work evaluates the similarity between individual original and reconstructed images. We, however, focus on measuring the similarity between the distributions of the original private dataset and the generated dataset.
>
> We will incorporate this paper, along with other relevant literature, into the related work section and clearly articulate how our approach differs.
>
> ### Weakness 3
>
> **W3:** It is unclear whether text-based representations are always sufficient to preserve fine-grained image details.
>
> **WA3:** We acknowledge that, for static evaluation, we cannot provide a definitive answer to whether text-based representations are always sufficient to preserve fine-grained image details. However, our approach demonstrates a clear improvement in preserving such details compared to existing methods. Furthermore, from a longer-term perspective, we believe this challenge will become increasingly addressable as foundation models continue to evolve. Our method is designed to seamlessly benefit from these advancements, allowing it to leverage future improvements in underlying models with minimal effort.
>
> ### Weakness 4
>
> **W4:** The baseline comparisons are incomplete.
>
> **WA4:** Thanks for the suggestion, we add the baseline as follows, and provide comparison between SPTI, PE Image and DP-finetune below. We fix the experiment settings as the same, with $\epsilon=1.0$, `num_samples=2000`. It's noticeable that we updated chat template for LLM, so we re-run all the previous experiments. Still, new results show that our method performs better than PE image and DP-finetune.
>
> Also, we will run more experiments with different settings to fully establish the excellence in our method.
>
> ||SPTI|PE Image| DP-finetune|
> |:-:|:-:|:-:|:-:|
> |LSUN Bedroom| **26.39** | 40.36 | 31.76 |
> |Wave-ui-25k| **37.39** | 50.45 | 62.08 |
> |European Art| **57.64** | 76.50 | 63.97 |
> |MM-Celeba-HQ| **34.17** | 50.45 | 57.01 |
>
> ### Weakness 5
>
> **W5:**  The DP guarantee is limited to the Private Evolution voting step, while other processes, particularly image-to-text captioning, access private data without DP protection. Although the paper cites the post-processing property of DP as justification, this design choice warrants further examination.
>
> **WA5:** PE process refers to the private data, and synthesize data that satisfies DP constraints. In our pipeline, we refers to captions, and generates image data that satisfies DP constraints, so it's fine for image-to-text captioning to access private data.
>
> ### Weakness 6
>
> **W6:** Computational costs are not fully discussed.
> **WA6:** We provide details of computational costs below in Question 4.
>
>
> ### Weakness 7
>
> **W7:** The paper does not fully analyze failure modes or limitations in the semantic fidelity of the generated images, especially when the text descriptions do not align well with the images in complex scene data.
>
> **WA7:** We test our method on various datasets to confirm that our pipeline works well in different circumstances, and experiments did show good results. However, we admit that there are cases where text description doesn't align well with the images. This usually happens when images have such intensive information that VLMs can hardly capture all of them, or limits on text length prohibits VLMs to fully describe all the scenes. Nevertheless, we believe the performance of our method will improve as state-of-the-art VLMs and diffusion models reach better results.
>
>
> ## Reply to Questions
> ### Question 1
> When using **text-voting** in our pipeline, the quality of generated caption **determines the upper bound** of DP generated data, since text samples are iteratively getting closer to private caption in text-voting. However, when using **image-voting** in our pipeline, text samples are directly referring to representation of private images, which actually doesn't need caption text for reference, so the overall performance is **not affected** by caption models.
>
> Notably, we uploaded a new chat template for the LLM we use and run experiments after paper submission, and found image-voting still performs better than text-voting even in Cat dataset, which argues that **image-voting performs generally better** than text-voting.
>
> However, as text-voting may be necessary in some condition, we run systematic experiments with text-voting with fixed settings. The following table shows the results. We will provide more comparison in our newest version of paper.
>
> | caption model  | LSUN Bedroom|
> |:----------:|:---------------:|
> |  gpt-4o-mini |  41.17   |
> | qwen-vl-max |   41.42  |
> | gemini-2.0-flash |  42.48 |
>
>
> ### Question 2
> It would work fine. Actually, many of the experiments conducted are using prompts like 'An image of ...', and received good results.
>
> As for privacy concerns, we assert that all the text prompts used in our experiments can be regarded as **public information**, which doesn't violate privacy. For example, if we consider LSUN Bedroom is a private dataset collected from sensitive user data, then 'this dataset is about bedroom' should be considered as public information, since the real sensitive part is the detail.
>
> ### Question 3
> Thanks for pointing out the deficiency in our work. We'll use **UMAP** for visualizing our results, and **CMMD** as an alternative metric other than **FID**. **CMMD** also show results better than **PE Image** method.
>
> **UMAP** shows our results align well with private data, and the visualization is uploaded in our newest version of paper. However, we can only provide our results in text, so only more results in **CMMD** is provided. In this table, we found our method performs the best except for the case in LSUN Bedroom. We suppose it's because LSUN Bedroom has such a large quantity of image data that improves the performance of fine-tuning, but this needs further experiments. Other conditions show the excellence of our method.
>
> | Dataset | SPTI | PE | DP-finetune |
> |:---:|:---:|:---:|:---:|
> | LSUN Bedroom | 1.744 | 2.066 | **1.376**  |
> | Wave-ui | **1.480** | 3.132 | 2.149 |
> | European_art | **0.684** | 1.016 | 2.447 |
>
> ### Question 4
> Our full data generation pipeline involves two major components: an 8B-parameter large language model (LLM) and a 3.5B-parameter diffusion model. Under our current implementation, a complete data generation run takes approximately 19 hours on **a single NVIDIA A800**, or 15.5 hours on **2×NVIDIA A100 GPUs**, with a peak memory requirement of about 70 GB.
>
> In comparison, the baseline method uses only a single 3.5B model, requires no LLM, and completes in about 7 hours on a **single NVIDIA A100 GPU**, with a peak memory requirement of about 35 GB. While our method has higher compute demands, it introduces a semantically enriched generation process through language-guided synthesis, which we show leads to superior data diversity and downstream performance.
>
> We believe the additional cost is justified by the significant gains in quality and generalization, and we note that our pipeline can be modularly optimized (e.g., via model distillation or prompt caching) in future work to reduce the runtime.
>
> As for API usage, caption process will take 250M tokens for 10,000 images. During data generation, LLM will process 7.3M tokens during each iteration, and diffusion model will process 5M tokens in each iteration. A total of 10 iterations will count for 123M tokens.

---

> > ### Comment · Reviewer_6vVq · 2025-08-05
> >
> > Thanks for answering my questions.
> > I still have a point of confusion regarding the answer to Q1 and would appreciate clarification.
> >
> > You state that image-voting “doesn’t need caption text for reference,” hence its performance should be unaffected by the captioner or prompt. However, after switching to a new chat-template, the relative ranking on Cat flipped in favour of image-voting. Could you clarify how a prompt change made prior to the image-voting step can impact the final FID, assuming that image voting is caption-independent?

---

> > > ### Author Response · Authors · 2025-08-06
> > > **Clarification on  Image Voting and Why Prompt of LLM Affects Image Voting Performance**
> > >
> > > Thank you for the thoughtful follow-up. We are happy to clarify.
> > >
> > > Although the image voting approach does not directly rely on a captioning model, the entire pipeline is driven by the text modality. Here is a short introduction of the image voting approach.
> > >
> > > 1. The synthesis process begins with Aug-PE. We use a LLM with **a proper prompt**, to generate an initial set of candidate texts.
> > >
> > > 2. The candidate texts are then iteratively refined through the Aug-PE process with VARIATION_API (**LLM API call**) and image voting. At the voting stage, each candidate text is used with an text2image model to generate an image, and image voting is performed by comparing these generated images with private images.
> > >
> > > Therefore, we can see that although image voting itself does not depend on captioning model, **the prompt and the LLM are used to generate initial candidate texts and the text variations**.  Improvements in the underlying LLM or prompts can affect the generated images and hence the final FID.
> > >
> > > We thank the reviewer again for highlighting this subtle but important point.

---

> > > > ### Comment · Reviewer_6vVq · 2025-08-06
> > > >
> > > > Thank you for the clarification. I will submit my final score after discussing and reviewing feedback from other reviewers.

---

### Note · Authors · 2025-08-13

Dear AC and reviewers,

Thank you for handling our paper and fostering a constructive and insightful discussion. Below are our final remarks summarizing the key points from the review process.



**1. Notable strengths identified in reviews**



*Novel ideas and modular framework.* Reviewers emphasized that using text as an intermediary for privacy-preserving image synthesis is both novel and practical (Deqn; ocKp). The proposed SPTI framework was regarded as a modular design and a meaningful technical advance (ocKp; 6vVq).

*Strong and extensive experiments.* The proposed methods show clear improvements over both Private Evolution and DP fine-tuning across multiple datasets and privacy budgets (Deqn; 6vVq; dX13).

*Clear presentation.* The paper  was commended for well-written and clearly presented (dX13).


**2. Key clarifications and additions during discussion**


* Added experiments directly comparing SPTI with PE and DP fine-tuning baselines, confirming consistent performance gains (Responses to ocKp; Deqn; 6vVq). Add additional evaluation metric CMMD, as requested by reviewers (Responses to ocKp; 6vVq), which also supports the consistent advantage of SPTI over various cases.

* Conducted a systematic study on the Cat dataset, revealing that dataset diversity significantly influences performance — SPTI favors more diverse private datasets (Response to 6vVq; Deqn; ocKp).

* Clarified the privacy analysis that our method strictly adheres to DP constraints (Response to Deqn).

* Clarified that SPTI does not rely on text captioning models but rely on LLMs for Random_API and Variation_API in the *image voting* case (Response to 6vVq; ocKp).

*These responses were well received by **all reviewers**, who expressed satisfaction with the rebuttal and clarifications provided.*

**3. Responses to ethical reviews**

We thank the Ethics Reviewers for their detailed feedback. Following recommendations from Ethics Reviewer uSHZ and AGFs, we will expand the discussion on potential negative societal impacts and misuse scenarios for privacy-preserving synthetic image generation.  We will add a dedicated statement in the “Broader Impact” section to acknowledge the risk that high-fidelity image synthesis could be misused for deceptive content, along with suggestions for responsible deployment.

*Once again, thank you for the thoughtful reviews and constructive exchange throughout the process, which have strengthened and refined our work.*

---

### Decision · Program_Chairs · 2025-09-17

**Decision:**

Accept (poster)

**Comment:**

This paper proposes Synthesis via Private Textual Intermediaries, a framework for privacy-preserving differentially private (DP) image synthesis. The method first captions private images using a VLM, then applies an APE algorithm to generate DP captions, and finally generates synthetic images from the DP text via a diffusion model. The entire pipeline is training-free and capable of producing high-resolution outputs. Experiments show favorable performance based on FID scores. Another contribution is a new dataset, Sprite Fright, for better benchmarking.

The strengths of the work lie in several aspects like interesting idea and sufficient novelty, not requiring model fine-tuning, modular pipeline, good writing, contributing a new dataset for the community, etc.

The main negative concerns include possible information loss via captioning, heavy dependence on the captioning capability of VLM, lack of baseline comparisons, sensitivity to changes in the image generation models, etc., most of which however have been resolved by the authors during rebuttal. Therefore, all the reviewers are finally satisfied with the work and give Borderline Accept (x4). Based on the above analysis, the AC think it deserves an Accept.